# T-TAMER:
# PROVABLY TAMING TRADE-OFFS IN ML SERVING

**Yuanyuan Yang**[*]
Department of Computer Science & Engineering
University of Washington

**Ruimin Zhang**
Department of Computer Science
University of Chicago

**Jamie Morgenstern**
Department of Computer Science & Engineering
University of Washington

**Haifeng Xu**[*]
Department of Computer Science
University of Chicago

## ABSTRACT

As machine learning models continue to grow in size and complexity, efficient serving faces increasingly broad trade-offs spanning accuracy, latency, resource usage, and other objectives. Multi-model serving further complicates these trade-offs; for example, in cascaded models, each early-exit decision balances latency reduction against potential accuracy loss. Despite the pervasiveness and importance of such trade-offs, current strategies remain largely heuristic and case-specific, limiting both their theoretical guarantees and general applicability.

We present a general framework, T-Tamer, which formalizes this setting as a multi-stage decision process, where the objective is to determine both when to exit and which model to consult. Our main result shows that recall (i.e., the ability to revisit earlier models) is both necessary and sufficient for achieving provable performance guarantees. In particular, we prove that strategies without recall cannot obtain any constant-factor approximation to the optimal trade-off, whereas recall-based strategies provably attain the optimal trade-off in polynomial time.

We validate our analysis through experiments on synthetic datasets and early-exit workloads for vision and NLP benchmarks. The results show that recall-based strategies consistently yield efficient accuracy–latency trade-offs. We hope this work provides a principled foundation for bridging heuristic practice with theoretical guarantees in the design of early-exit and cascaded models.

## 1 INTRODUCTION

As models continue to grow in scale, relying on a single model often fails to meet all service-level objectives (SLOs), such as accuracy, latency, and cost. Deploying only the largest model for every query is both impractical and suboptimal for inference platforms, as many queries can be effectively handled without resorting to the most resource-intensive option (Nie et al., 2024; Rahmath P et al., 2024; Matsubara et al., 2022).

Motivated by this observation, *cascaded inference* has emerged as a widely adopted paradigm for efficient large-scale model serving. The central idea is to maintain a collection of sub-models with varying complexity and to invoke them adaptively in sequence. In practice, the inference platform receives a stream of queries for a classification task together with specified SLOs, such as achieving high accuracy under an average latency budget. Cascaded inference routes simple queries to lightweight models, while reserving the most complex models for the hardest cases. By adaptively selecting sub-models based on query characteristics, the up-cascade framework provides a principled mechanism for efficient inference in modern machine learning systems.

Alternatively, the inference platform can be viewed as taming the trade-offs among conflicting SLOs. While the accuracy–latency trade-off is the most prominent, similar tensions arise in ac-

---

[*]Corresponding author.

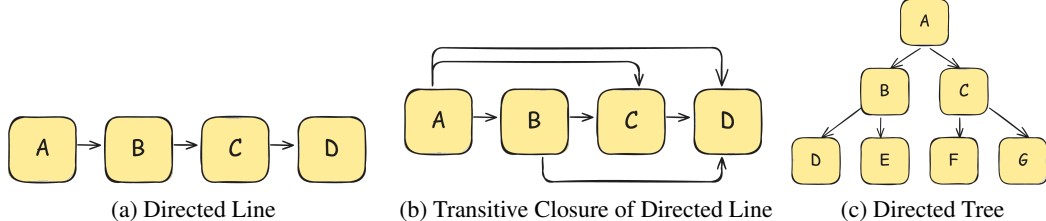

(a) Directed Line     (b) Transitive Closure of Directed Line     (c) Directed Tree

Figure 1: **DAG structures considered in this work: the directed line, its transitive closure, and the directed tree, which capture common topologies in up-cascade inference.**

curacy–cost (where more accurate models generally incur higher computational/monetary cost) and latency–throughput (where larger batch sizes improve throughput but increase per-sample latency).

However, there is no *universally accepted* policy for sub-model routing and termination that applies across different trade-offs and use cases. Existing policies are typically developed in an *ad hoc* manner and remain largely *heuristic-driven*.[1] This gives rise to two fundamental issues: (1) *limited generalizability*, as a policy designed for one use case often fails to transfer to others, and (2) *suboptimality*, as the resulting strategy is either inefficient or lacks provable efficiency guarantees.

We propose T-Tamer, a general theoretical framework for taming bi-objective trade-offs in cascaded inference. At its core, the framework computes a *theoretically optimal* strategy and instantiates it as a data-driven learner that fits this solution using input–output pairs from *all* sub-models. At inference time, given a query, T-Tamer incrementally updates its belief over sub-model performance as models are inspected and computes the optimal routing and termination policy based on this belief. Notably, the training of the T-Tamer is *agnostic* to that of the sub-models, enabling it to operate as a plug-in component rather than a case-specific solution.

The key to T-Tamer strategy is a theoretical abstraction of routing and termination as a multi-stage costly exploration defined over a directed acyclic graph (DAG), where nodes represent sub-models and edges represent both *precedence constraints* (i.e., model A must precede model B) and performance dependencies (i.e., the output of model B conditionally depends on the outputs of model A). The policies derived by T-Tamer are supported by our theoretical analysis and can be computed in polynomial time for DAG structures that commonly arise in practice (See Fig. 1): the directed line, its transitive closure (allowing any skip while preserving order), and the directed tree (decision-tree structure).

## 1.1 Applications and Prior Work

The DAG topologies studied here naturally arise in practical inference systems. We review models and prior work that instantiate these structures and connect them to research on multi-model inference, cascaded architectures, and adaptive computation. See App. A for a complete discussion.

**Intra-Model Cascaded Inference**. Existing intra-model inference methods manage trade-offs by adaptively determining whether to incorporate deeper layers of the network during inference (see surveys by (Han et al., 2021; Montello et al., 2025)). Since the layered architecture of neural networks naturally induces Markovian dependencies, these approaches can be regarded as special cases of the cascaded inference framework. Representative examples include early-exit models (Teerapittayanon et al., 2016; Xin et al., 2020; Matsubara et al., 2022; Rahmath P et al., 2024), skip networks (Wang et al., 2018), and dynamic recursive neural networks (Guo et al., 2019).

**Inter-Model Cascaded Inference**. More recently, inter-model cascaded inference has been adapted to large models (including LLMs), to mitigate their high inference cost. Existing works have explored how to jointly train the cascades (Varshney & Baral, 2022), how to design cascades with awareness of system-level constraints (Lebovitz et al., 2023) and how to leverage signals to route the model (Varshney & Baral, 2022). These efforts show the promise of cascaded inference for large-scale model serving, yet the absence of guarantees exposes a fundamental theoretical gap.

---

[1] As an aside, exhaustive search is prohibitively time-consuming in most use cases.

## 1.2 OUR RESULTS

We formulate the trade-off between the two objectives as a weighted sum of their proxy loss functions, with weights controlled by a tunable parameter $\lambda$. More concretely, probing and consulting exactly one model on input $x$ corresponds to the objective

$$\theta_\lambda(x) = \lambda\ell_1(x) + (1 - \lambda)\ell_2(x),$$

where $\lambda \in [0, 1]$ and $\ell_1, \ell_2$ are loss functions depending on $x$. This additive formulation is (i) **flexible**, as it can directly incorporate standard loss functions used in machine learning models, and (ii) **stable**, since $\theta_\lambda(x)$ is always well-defined, whereas subtraction-based formulations may produce pathological behavior such as negative or unbounded objectives.

Building on this formulation, we note that because inputs are drawn from a distribution, the induced losses of sub-models follow a joint correlated distribution. Our T-Tamer captures this structure by assigning losses to edges and nodes, thereby reducing routing, stopping, and consulting in cascaded models to a *costly exploration* problem over a directed acyclic graph (DAG) that encodes both precedence constraints and inter-model correlations:

- **Optimal Stopping Under Single Line Setting**(§4): We classify strategies based on whether they allow recall, that is, whether previously explored sub-models can be revisited and selected after further exploration. Specifically, we develop separate theoretical frameworks for no-recall and with-recall strategies in the single-line setting. We show that no-recall strategies fail to achieve any constant-factor approximation to the optimal utility (§3). This impossibility is *information-theoretic*, rather than computational, and thus cannot be circumvented by increased computational resources. Notably, existing confidence-based cascaded models (Xin et al., 2020; Teerapittayanon et al., 2016; Laskaridis et al., 2021) can be viewed as no-recall strategies, and our results therefore reveal the fundamental limitations of such heuristics.

  For the with-recall setting, we establish a provably optimal *dynamic indexing* strategy. At each step, the strategy computes the index of the next available model and decides whether to stop or continue based on this value. Upon stopping, it returns the best model among those inspected. Moreover, this strategy can be computed and implemented efficiently via dynamic programming.

- **General Costly Exploration for More General DAGs**(§5): Our work generalizes the dynamic indexing strategy beyond the single-line setting, where at every iteration, the strategy computes the indices of *all* available models that remain uninspected. We prove that for both tree topologies (Fig. 1c) and the transitive closure of the directed-line case (Fig. 1b), this indexing strategy remains theoretically optimal and can be computed in polynomial time.

Finally, we evaluate our dynamic indexing strategy on synthetic datasets and early-exit workloads from standard vision and NLP benchmarks (§6). The results show that recall-based strategies consistently deliver efficient accuracy–latency trade-offs. We now present our main results:

- **General-Purpose Tradeoff Tamer for Up-Cascade Inference**. We introduce T-Tamer, a principled framework for taming bi-objective trade-offs in cascaded inference. Unlike heuristic-driven methods, T-Tamer is model-agnostic and learns to compute the theoretically optimal routing and stopping policy across all sub-models. This makes it broadly applicable as a plug-in component for diverse cascaded inference systems without requiring case-specific tuning.

- **No Constant-Factor Approximation for No-Recall Policy**. We prove an *information-theoretic* impossibility for no-recall strategies, showing that even in the directed-line case, such policies cannot achieve any constant-factor approximation to the optimal utility. This result shows that confidence-based heuristics (i.e., early-exit thresholds) for cascaded inference are inherently suboptimal, motivating recall-based strategies with stronger guarantees.

- **Provably Efficient Routing-and-Stopping Policy over DAGs**. We develop a *dynamic indexing* strategy that provably achieves optimal exploration and stopping across canonical DAG structures arising in cascaded inference (directed lines, their transitive closures, and trees). Our approach is not only theoretically optimal but also computationally efficient, running in polynomial time. This bridges the gap between theory and practice by providing the first provable, efficient policies for real-world cascaded inference topologies.

## 2 PROBLEM FORMULATION

In this section, we establish the notation, assumptions, and formal framework of costly exploration over general DAGs, which serves as the theoretical model for cascaded inference.

**Notations**. Throughout, let $n$ denote the number of sub-models in the cascade, and let $\mathcal{X}$ denote the input space with $x \in \mathcal{X}$. We use $T$ to denote the number of samples used to fit the costly inspection strategy. We use $\ell$ to denote the dominant loss function and $c$ to denote the secondary loss function, occasionally abusing notation by referring to $c$ as the (inspection) cost.

We impose two standard assumptions in supervised learning (Goodfellow et al., 2016). First, all losses are strictly positive[2]. Second, inputs are drawn from a distribution.

**Assumption 2.1** (Loss). *For any sub-model $j \in [n]$ and input $x \in \mathcal{X}$, the losses satisfy $\ell_j(x) > 0$ and $c_j > 0$. Moreover, the cost loss $c_j$ is a constant independent of the input.*

**Assumption 2.2** (Distributional Assumption). *Each $x_t$, for sample index $t \in [T]$, is assumed to be i.i.d. from a fixed distribution $\mathcal{D}$ over $\mathbb{R}^d$.*

We distinguish between *recall* and *no-recall* strategies, depending on whether previously consulted models remain available as candidates.

**Definition 2.3** (Recall / No-Recall Strategy). *Given a policy that terminates after consulting sub-model $i \in [n]$, the strategy is **no-recall**, if the final prediction must come from sub-model $i$; **with-recall**, if the policy may return the prediction of any sub-model $j \leq i$.*

We formalize the costly exploration problem over a DAG. Specifically, we introduce a dummy root node $v_0$ as the starting node of the policy, in addition to the nodes representing sub-models.

**Problem 2.4** (Markovian Costly Exploration). *Consider a set of $n$ nodes $\mathcal{V} = \{v_1, \ldots, v_n\}$. Each node $v_i$ is associated with a random loss $\ell_i$ drawn from a known distribution $\mathcal{D}_i$. We also include a designated null node $v_0$, which connects to $v_1$ but to no other node, with $\ell_0 = 0$. The nodes are organized into a directed acyclic graph $G = (\mathcal{V}, E)$, where edges encode:*

- ***Partial ordering**. For any edge $(v_i, v_j) \in E$, if node $v_j$ is probed, then $v_i$ must be probed strictly before $v_j$. We denote this as $v_i \prec v_j$. Note that $v_i$ itself need not be probed unless $v_j$ is selected.*
- ***Edge cost**. For each edge $(v_i, v_j) \in E$, probing $v_j$ right after $v_i$ incurs an edge cost $c(i, j)$.*
- ***Markov property**. For any adjacent nodes $v_i \prec v_j \prec v_k$ along a directed path in $G$, the node losses satisfy the conditional independence $\ell_i \perp \ell_k \mid \ell_j$.*

*A policy $\pi$ starts from $v_0$ and adaptively selects edges to probe subsequent nodes, deciding at each step whether to continue or stop. Let $\mathcal{O}(\pi)$ denote the set of probed nodes and $E(\pi)$ the set of edges traversed by $\pi$. The goal is to minimize the sum of node and edge losses weighted by the tradeoff parameter $\lambda$: $\mathbb{E}[\lambda \cdot f([\ell_i]_{i \in \mathcal{O}(\pi)}) + (1 - \lambda) \cdot \sum_{e \in E(\pi)} c(e)]$.*

*For no-recall, $f$ equals the loss of the last visited node; for with-recall, $f$ equals $\min_{i \in \mathcal{O}(\pi)} \ell_{1,i}$.*

Finally, we describe how a general policy routes through the sub-models in the cascade (Figure 2).

---

For each input $x \in \mathcal{X}_x$:

1. The policy observes the input $x$ and sets $i \leftarrow 1$.

2. The policy queries sub-model $i$ and observes its loss $\lambda \ell_i(x)$.

3. The policy decides whether to stop or continue:

   - **Continue**: Among the sub-models available after sub-model $i$, choose the next sub-model $j$ to probe, incur edge cost $(1 - \lambda)c(i, j)$ and return to Step 2.
   - **Stop**: Return the prediction of a selected sub-model.

---

Figure 2: **Costly Exploration for Cascaded Inference**

---

[2]Loss functions that may take negative values can be shifted or transformed to satisfy this condition.

# 3 ON COSTLY EXPLORATION POLICIES WITH NO RECALLS

In this section, we establish an information-theoretic bound showing that no-recall strategies cannot achieve a constant-factor approximation to the offline optimal loss, even in the single-line setting.

## 3.1 CONNECTION TO SEQUENTIAL CASCADED INFERENCE

The single line case corresponds to the sequential cascaded inference paradigm widely adopted in practice. In sequential CI, sub-models are arranged in a *fixed* order and must be inspected one by one (Xin et al., 2020; Dekoninck et al., 2024). A no-recall strategy in this setting amounts to always serving the most complex model inspected so far. A prominent approach is the confidence-threshold strategy, which stops and serves the current model once its prediction confidence exceeds a predefined threshold (Dai et al., 2024; Xin et al., 2020).

## 3.2 EXIT WITH NO RECALL: NO CONSTANT APPROXIMATION

In the single-line case, the costly no-recall exploration problem can be formalized as an optimal stopping problem over a sequence of losses $R_i$ with Markovian dependencies. Specifically, $R_{i+1} \sim \lambda \ell_i + (1 - \lambda)c(i - 1, i)$ for $i \in [n]$.

**Problem 3.1** (No Recall Exit Problem). *Let costs $R_1 \sim \mathcal{D}_1, \ldots, R_n \sim \mathcal{D}_n$ be non-negative random variables drawn from known distributions $\mathcal{D}_1, \ldots, \mathcal{D}_n$, with a joint distribution exhibiting Markovian dependency, i.e., for all $i \in [n]$, $R_{i+1}$ is conditionally independent of the past given $R_i$:*

$$\Pr(R_{i+1} \mid R_1, \ldots, R_i) = \Pr(R_{i+1} \mid R_i), \quad \textit{for all } i.$$

*A decision maker sequentially observes $R_1$ to $R_n$. After observing $R_i$, they must either stop and pay the cost $R_i$ or irrevocably discard and continue. The goal is to design a stopping rule ALG that minimizes the expected loss.*

Ideally, ALG is benchmarked against the optimal loss attainable with *perfect* knowledge of all $R_i$.

**Definition 3.2** (Offline Optimal). *The benchmark is an oracle who knows all realizations in advance and selects $\min_i R_i$, incurring expected cost $\mathrm{OPT} = \mathbb{E}[\min_i R_i]$.*

As achieving the offline optimal loss is generally infeasible without full future information, the best attainable guarantee lies in bounding the approximation ratio.

**Definition 3.3** (Approximation Ratio). *We say ALG is an $\alpha$-approximation if for some $\alpha \geq 1$,*

$$\mathbb{E}[\mathrm{ALG}] \leq \alpha \cdot \mathrm{OPT}.$$

We concluded by showing that no algorithm can achieve a constant approximation ratio for no recall costly exploration problem, even when the underlying distribution is bounded.

**Theorem 3.4** (Impossibility of Constant Approximation, No-Recall Costly Exploration). *For no-recall costly exploration problem (Prob. 3.1), no algorithm achieves a bounded $\alpha$-approximation ratio, even with $n = 2$ and bounded distributions.*

*Proof Sketch.* Let $n = 2$ and $\alpha > 1$ be an arbitrary large constant. Consider the following random variables:

$$R_1 = \frac{1}{\alpha^2} \qquad \text{w.p. } 1, \qquad R_2 = \begin{cases} 0 & \text{w.p.} 1 - \frac{1}{\alpha}, \\ \frac{1}{\alpha} & \text{w.p. } \frac{1}{\alpha} \end{cases}$$

Under this construction, any algorithm achieves an expected reward of exactly $1/\alpha^2$, but a prophet achieves $\mathrm{OPT} = 1/\alpha^3$. This indicates an $\alpha$-competitive ratio. This competitive ratio can be made arbitrarily large by increasing $\alpha$. □

This theorem establishes an *information-theoretic* impossibility: no no-recall costly exploration policy can achieve any non-trivial approximation to the offline optimum. Importantly, this limitation is not due to computational hardness (e.g., NP-hardness), but stems from the intrinsic information structure of the problem. One might ask whether it is possible to design an algorithm ALG that approximates a restricted class of benchmarks. However, such benchmarks can be trivial to approximate—for example, in the construction above, all strategies achieve the same expected utility.

## 4 WARM-UP: INDEXING OVER A DIRECTED LINE

While the no-recall approach is natural, it overlooks an important practical phenomenon: larger models are not always superior to smaller ones and may even "over-think," producing worse predictions than intermediate models (Sui et al., 2025; Kaya et al., 2019). Such behavior has been observed in real systems, underscoring the practical need for recall-based strategies that can revisit earlier models and provide stronger guarantees. In this section, we introduce a theoretically optimal indexing strategy for with-recall costly exploration for single-line setting.

### 4.1 WITH-RECALL COSTLY INSPECTION OVER SINGLE LINE

Motivated by the theoretical limitations of no-recall policies discussed earlier, we now turn to analyzing the efficiency of with-recall policies. The with-recall setting admits a similar abstraction under costly exploration. The key distinction from the no-recall formulation is that the two losses cannot be collapsed into a single objective: one loss $R_i = \lambda \ell_i$ is incurred at the nodes, while a separate cost $c_i := (1 - \lambda)c(i - 1, i)$ is incurred along the edges when moving to the current node.

**Problem 4.1** (With-Recall Costly Exploration). *Let the costs of the nodes $R_1, \ldots, R_n$ be non-negative random variables drawn from known distributions $\mathcal{D}_1, \ldots, \mathcal{D}_n$, with a joint distribution exhibiting Markovian dependency; i.e., $R_{i+1}$ is conditionally independent of the past given $R_i$.*

*A decision maker sequentially observes $R_1, \ldots, R_n$, where each node $i$ incurs a cost $\lambda c_i$. After observing $R_i$, the decision maker must either stop—incurring a total cost of $\min_{k \in [i]} R_k + \sum_{j \in [i]} c_j$ by selecting $\arg\min_{k \in [i]} R_k$—or continue. The goal is to minimize the expected total cost.*

Since Markovian-correlated distributions with continuous support cannot be directly represented without additional assumptions (Ethier & Kurtz, 2009), we quantize them into a discrete domain and base decisions on this discretization. Such discretization is standard in practice (e.g., grid search). Hence, without loss of generality, we assume $\mathcal{D}_1, \ldots, \mathcal{D}_n$ are discrete.

Note that the counterexample in Theorem 3.4 continues to yield arbitrarily large approximation gaps even when recall is allowed. We therefore benchmark against a more favorable comparator with tractable guarantees.

**Definition 4.2** (Online Optimal). *The benchmark is the optimal online algorithm, which has access to the joint distribution $\mathcal{D}_1, \ldots, \mathcal{D}_n$ and achieves the minimum possible expected loss without observing realizations in advance.*

By Bellman's principle of optimality, we derive the optimal with-recall costly exploration strategy via dynamic programming, starting from the last node and progressively extending to the first. This yields a clean structural result: the optimal policy stops once the current minimum loss falls below a dynamic index $\sigma$ determined by the current observation.

---

**Algorithm 1** Costly Exploration via Indexing

---

**Require:** Nodes $\{v_1, \ldots, v_n\}$, edge costs $\{c_1, \ldots, c_n\}$, dynamic index $\sigma(i, s)$ (Def. 4.4) for all $i$ and $s \in V$.
1: Initialize minimum loss $X \leftarrow \infty$, $i \leftarrow 1$, $\sigma \leftarrow \sigma(1, \emptyset)$.
2: **while** $X > \sigma$ **do**
3:      Pay $c_i$ to inspect node $v_i$, observe loss $R_i$.
4:      $X \leftarrow \min\{X, R_i\}$.                      ▷ Update minimum loss.
5:      $\sigma \leftarrow \sigma(i + 1, R_i)$.                        ▷ Update threshold.
6:      $i \leftarrow i + 1$.
7: **end while**
8: **Return** node $v_j$ that has been inspected with the minimum loss.

---

Algorithm 1 illustrates the structure of the optimal strategy. At each step, the algorithm updates the dynamic index of the next box and decides whether to stop given the current minimum loss. Thus, the stopping decision at step $i$ can be represented as a stop/continue rule table dependent on $(X, R_i)$.

## 4.2 THE DYNAMIC INDEX

We now introduce the dynamic index, which serves as the core of our provably efficient strategy. At a high level, the optimal policy stores, for every possible state $(X, R_{i-1}, i)$, the corresponding index and the stop/continue decision. These indices can be computed efficiently via dynamic programming, proceeding backward from the last node.

**Notation.** Assume that each $R_i$ takes values in a common finite support $V = \{v_1, \ldots, v_k\}$. For each $i \in [n]$, let $\mathbf{p}_i$ denote the probability mass function (PMF) of $R_i$, with $\mathbf{p}_i[v_q] = \Pr[R_i = v_q]$. Let $P_i \in \mathbb{R}_+^{k \times k}$ be the transition matrix from $\mathcal{D}_i$ to $\mathcal{D}_{i+1}$, such that $\mathbf{p}_{i+1} = \mathbf{p}_i \cdot P_{i+1}$. The optimal policy $\pi$ reduces to a stopping rule that selects the node with minimum loss, we use $\tau$ to denote it.

We now formally describe the dynamic programming procedure to compute the dynamic index function $\sigma$. Any stopping time $\tau$ depends only on the current minimum reward $X$, the most recent loss $R_i$, and the next candidate index $i + 1$. We refer to the tuple $(X, R_i, i+1)$ as the algorithm's *state*, and proceed to derive the expected loss incurred at this state under a given stopping time $\tau$.

**Definition 4.3** (Equivalent Loss). *Given $\tau$ and $(x, R_{i-1}, i)$, we define the expected loss of the state $(X, R_{i-1}, i)$ following stopping rule $\tau$ as $\Phi^\tau(X, R_{i-1}, i)$.*

$$\Phi^\tau(X, R_{i-1}, i) := \mathbb{E}[\min\{X, \min_{j=i}^{\tau(X, R_{i-1}, i)} R_j\} + \sum_{j=i}^{\tau(X, R_{i-1}, i)} c_j]$$

*In addition, we use $\Phi(X, R_{i-1}, i) = \Phi^{\tau^*}(X, R_{i-1}, i) = \min_\tau \Phi^\tau(X, R_{i-1}, i)$ to denote the expected future loss following the optimal strategy $\tau^*$ starting at state $(X, R_{i-1}, i)$.*

Let $\Phi$ be the expected loss of a state, then $\Phi$ can be solved inductively using Bellman's principle of optimality: $\Phi^{\tau^*}(X, R_{i-1}, i) = \min\left\{X, c_i + \mathbb{E}_{R_i|R_{i-1}}[\phi^{\tau^*}(\min\{X, R_i\}, R_i, i + 1)]\right\}$, where the first term corresponds to stopping immediately, and the second to continuing and applying the optimal stopping time $\tau^*$ for the remaining boxes.

Importantly, for fixed $R_{i-1}$ and $i$, there exists a maximal $X$ such that the decision maker is *indifferent* between stopping and opening the next box. This value defines the dynamic index $\sigma$ that governs our algorithm (Alg. 1). More formally,

**Definition 4.4** (Dynamic Index). *Given any state $(X, R_{i-1}, i)$, we define the dynamic index at the current state, denoted by $\sigma_i(X, R_{i-1}, i)$, as the smallest solution to:*

$$\mathbb{E}\left[\left(\sigma - \min_{j=i}^{\tau^*(\sigma, R_{i-1}, i)} R_j\right)_+ - \sum_{j=i}^{\tau^*(\sigma, R_{i-1}, i)} c_j\right] = 0, \tag{1}$$

*where $\tau^*$ is the optimal strategy.*

## 4.3 PROVABLE EFFICIENCY

Next, we establish that the dynamic index is both well-defined and optimal, thereby justifying the correctness of our algorithm (Alg. 1). Moreover, we show that the optimal strategy can be computed efficiently via dynamic programming. See Sec. B for more details.

**Theorem 4.5** (Optimality and Efficiency of Dynamic Indexing). *Given the current state $(X, R_i, i+1)$, there exists a solution to (1). This solution is independent of the current minimum loss $X$ and can be denoted by $\sigma(R_i, i+1)$. The indexing policy that stops when $\sigma > X$ and continues otherwise (Alg. 1) achieves online optimality.*

*Furthermore, preprocessing this policy takes $O(n \cdot |V|^2 T)$ time and requires $O(n|V|^2)$ space. At inference time, for each input $x$, the policy runs in $O(1)$ per node and $O(n)$ overall per input.*

The intuition behind the guarantee is that the stop/continue decision for each state $(X, R_i, i+1)$ can be precomputed and stored. At inference time, the algorithm queries this table in $O(1)$ per step, yielding $O(n)$ time per sample. Further details are given in Lem. B.3 and Lem. B.4.

*Remark.* In the single directed line setting, no routing decisions are required—each step involves only a binary choice of whether to proceed or not. Consequently, by Bellman's principle of optimality, our indexing strategy remains optimal under arbitrary correlation structures.

## 5 EXTENSION: STRATEGIES OVER GENERAL DAGS

In this section, we extend the dynamic index to general DAGs. The generalized indexing policy remains optimal, but it must additionally incorporate the routing decision, i.e., which model to inspect next. We show that this policy can still be implemented efficiently via dynamic programming.

### 5.1 DIRECTED TREE

We describe how to generalize the previous indexing strategy to the directed tree setting. One application of this setting is cost-aware binary search over domains, where the tree corresponds to a binary search tree and the loss represents the cost of acquiring feedback, as in human-in-the-loop settings such as RLHF (Xiong et al., 2023) or crowdsourcing.

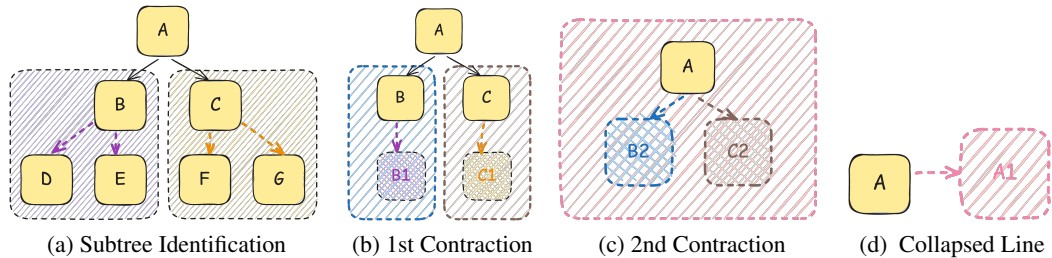

(a) Subtree Identification     (b) 1st Contraction     (c) 2nd Contraction     (d) Collapsed Line

Figure 3: **An Illustration on Node Contraction**

Illustrated by Fig. 3, the key idea of our generalized dynamic index is to define a tree contraction procedure that contracts a subtree into a single node while preserving the equivalent loss table and loss distribution of the subtree. Concretely, we first identify subtrees whose children consist only of single nodes or multi-lines, contract these into single nodes, and then repeat the process iteratively.

To carry out this indexing policy, we can still keep an if-stop matrix for each node, which only depends on the realized minimum loss and the loss of that node. The main difference is that now, when using dynamic programming, we need to combine information from all of the node's children.

**Theorem 5.1** (Dynamic Indexing in Directed Trees: Optimality and Efficiency). *There exist a generalization of the dynamic indexing policy (Alg. 3), which is theoretically optimal. Furthermore, preprocessing this policy takes $O(n \cdot |V|^2 T)$ time and requires $O(n|V|^2)$ space. At inference time, for each input $x$, the policy runs in $O(1)$ per node and $O(n)$ overall per input.*

We defer the details to Lem. C.13 and Thm. C.14 in the Appendix C.

### 5.2 TRANSITIVE CLOSURE OF A DIRECTED LINE

We extend the dynamic index to the transitive closure of a directed line. Unlike the directed line setting, where models must be evaluated strictly sequentially, the transitive closure allows skipping while preserving order, enabling cost savings by reducing the number of evaluations.

**Theorem 5.2** (Dynamic Indexing in Skipped Inference: Optimality and Efficiency). *There exists a generalization of the dynamic indexing policy, which is theoretically optimal. Furthermore, preprocessing this policy takes $O(n^2 \cdot |V|^2 T)$ time and requires $O(n|V|^2)$ space. At inference time, for each input $x$, the policy runs in $O(1)$ per node and $O(n)$ overall per input.*

We can still use dynamic programming to pre-compute the if-stop matrix and the equivalent-loss table. The key difference from the single-line case is that, when computing the equivalent loss, we must enumerate over all possible next nodes rather than just the immediate successor. This increases the preprocessing time by a factor of $n$. We defer the technical details to Section C.3.

# 6 EXPERIMENTS

In this section, we evaluate our dynamic indexing strategy RECALL over real-world CV/ NLP early exit (EE) classification workloads. Early exit workloads are selected as the primary experimental setting, corresponding to a *dynamic indexing* strategy over a single directed line. The evaluation includes standard computationally efficient offline early exit benchmarks, which do not update their policies at runtime. The results demonstrate the generality of T-tamer, showing that it consistently matches or exceeds the performance of the benchmark algorithms, particularly in high accuracy regimes, even though the best performing benchmark varies across workloads.

More details are in Appendix D.

## 6.1 IMPLEMENTATION DETAILS

**Metrics**. We use the error rate as a metric, defined as $\text{Err} = 1 - \text{Acc}$, where Acc is the empirical accuracy measured against the outputs of the backbone model, which we regard as an upper bound on achievable performance given the model's capacity. To demonstrate latency reduction, we *normalize* the achieved latency against the original latency.

**Benchmarks**. We compare the following baselines in our experiments:

(i) **Score-based methods**, where EE inference terminates once a sub-model's score exceeds or falls below a specified threshold. In our experiments, we adopt a confidence-based approach (Xin et al., 2021), in which confidence—measured as the softmax probability of the predicted class—triggers an exit when it surpasses a predefined threshold. Since the latency of this algorithm strictly decreases as the loss threshold increases, we can use binary search to find the threshold that meets a given latency requirement.

(ii) **Rule-based methods**, in which EE termination is governed by heuristic criteria. We adopt a theoretically motivated variant—patience-based methods (Zhou et al., 2020)—which trigger an exit once $K$ consecutive sub-models produce sufficiently similar predictions. This consistency is measured by the change in predictive confidence/loss between adjacent sub-models: If this change is below a threshold $\tau$, the stability counter $C$ is incremented. If $C$ reaches $K$ during inference, the model exits at the current sub-model; otherwise, it exits at the final sub-model. In our experiments, we set $K = 2$, to obtain the fastest possible early exit.

## 6.2 EVALUATIONS

**Pareto Frontier**. In Figure 4, we evaluate our dynamic index strategy over vision classification tasks, using the video streaming dataset collected from (Agarwal & Netravali, 2023; Hsieh et al., 2018), with VGG-{11, 13} models (Simonyan & Zisserman, 2014) as the backbone models for EE. Our method attains approximately 95% accuracy with about 50% latency savings on CV workloads, and achieves around 75% accuracy with roughly 90% latency savings on NLP workloads. For detailed experimental settings, we defer to Appendix D.3, and for additional NLP workload results, we defer to Appendix D.4.

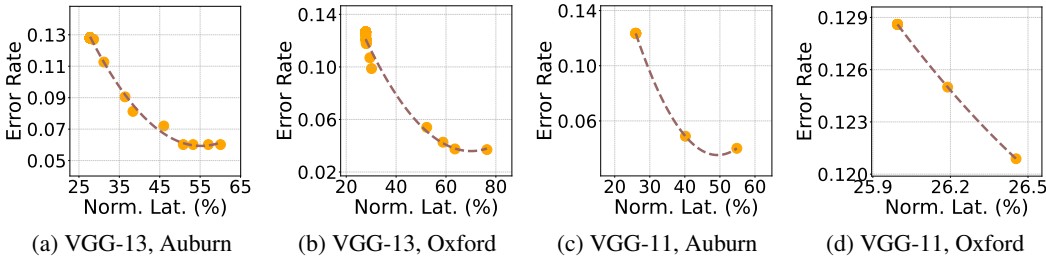

| (a) VGG-13, Auburn | (b) VGG-13, Oxford | (c) VGG-11, Auburn | (d) VGG-11, Oxford |

Figure 4: **Pareto Frontiers for Vision Models.** The frontiers highlight regions where latency is significantly reduced with only limited accuracy degradation. For instance, Fig. 4a shows that latency is reduced to 45% of the original, while sacrificing less than 7% accuracy.

**Comparison with Benchmark Models**. Figure 5 shows that T-Tamer consistently identifies the best points in the high-accuracy region relative to other benchmarks, while remaining competitive with the 1-threshold baseline for GPT-2 models and staying within 10% for BERT-based models in lower-accuracy regimes. It also consistently outperforms patience-based methods across NLP workloads. Notably, for BERT-based workloads, the Pareto front produced by 1-thresholding violates monotonicity, indicating that *deeper exits are not uniformly beneficial*.

In such cases, adding recall provides a more effective signal, which aligns directly with our motivation for designing t-tamer as a recall-driven strategy.

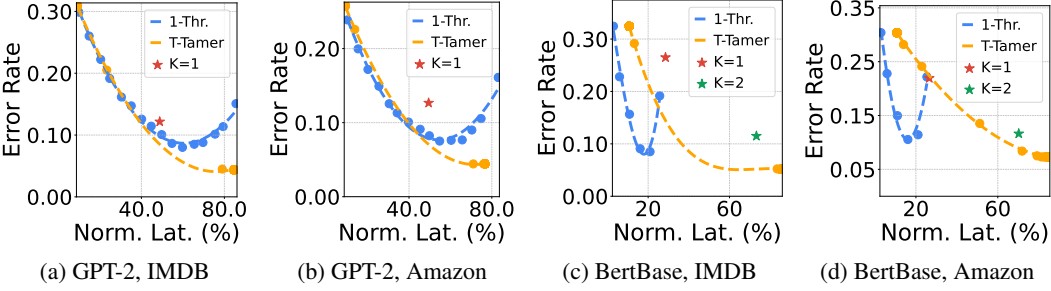

(a) GPT-2, IMDB    (b) GPT-2, Amazon    (c) BertBase, IMDB    (d) BertBase, Amazon

Figure 5: **Benchmark Comparison**. The plot reports T-Tamer accuracy on the NLP tasks with additive discretization 0.001. X-axis: Normalized Latency of the backbone model; Y-axis: Error.

**SSA Approximation Quality**. Next, we present experimental results using soft state aggregation (SSA)(Duan et al., 2019) based T-TAMER, introduced to accelerate its matrix-multiplication step. The SSA parameter is the number of meta-states $q$. Under a fixed latency requirement, we ran T-TAMER for $q \in 50, 100, \dots, 800$. We find that once $q > 100$, the accuracy remains within $\pm 0.02$ of that achieved with the exact transition matrix. Details are deferred to App. D.2.

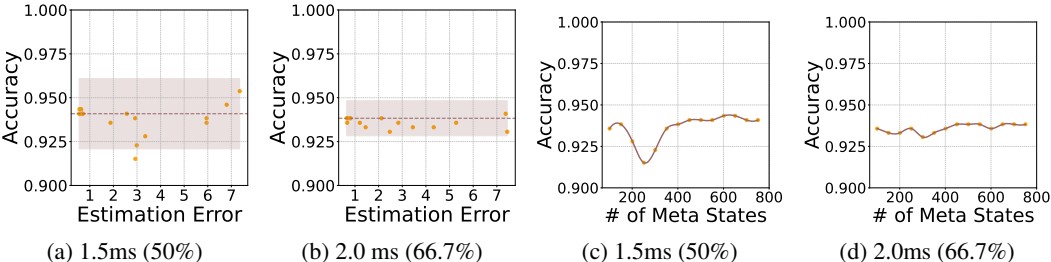

(a) 1.5ms (50%)    (b) 2.0 ms (66.7%)    (c) 1.5ms (50%)    (d) 2.0ms (66.7%)

Figure 6: **Effect of SSA Approximation Granularity on Accuracy**. The plot reports T-Tamer accuracy on the Auburn dataset using a VGG-13 backbone with additive discretization 0.001. The horizontal line gives the true accuracy, and the shaded band shows the $\pm 0.02$ margin. X-axis: # of Meta-states; Y-axis: Accuracy. Each caption indicates the latency in ms (% of time saved).

Figure 6 illustrates the dependence of accuracy on the number of meta-states and on the estimation error. The performance remains largely stable across meta-state configurations. When the number of meta states reaches approximately 300, the SSA-based T-Tamer closely matches the performance of vanilla T-Tamer using the true transition matrices (i.e., up to 0.02%). This suggests that the SSA formulation provides an efficient surrogate for regimes with large transition matrices (i.e., $1000 \times 1000$ in our setting).

# 7 CONCLUSION

We introduced T-Tamer, a principled framework for taming bi-objective trade-offs in cascaded inference. By formulating routing and stopping as a costly exploration problem over DAGs, we developed a dynamic indexing strategy that achieves provable optimality with polynomial-time complexity. Experiments on synthetic data and real-world CV/NLP benchmarks confirm that T-Tamer consistently delivers generality and favorable accuracy–latency trade-offs, providing a general-purpose and efficient foundation for modern inference platforms.

## ACKNOWLEDGEMENT

We sincerely thank the anonymous reviewers of ICLR 2026 for their thoughtful feedback and constructive suggestions, which motivated us to incorporate soft state aggregation methods and additional experiments, thereby substantially improving this work. This project was supported by NSF CCF-2045402 and NSF CCF-2019844.

A theoretical, TCS-oriented version of this work appears on arXiv as *Markovian Pandora's Box* and is released solely as a technical report (not under submission). In contrast, this paper emphasizes the application and ML systems perspective and studies the minimization formulation of Pandora's box, whereas the arXiv version considers the maximization setting. An earlier version appeared as *Hold That Exit: Near Optimal Early-Exit Inference via Recall* at the NeurIPS SPIGM 2025 Workshop.

YY gratefully acknowledges helpful discussions with Linda Cai, Li Chen, and Shuchi Chawla. YY also notes that portions of the system-level insights in this work were inspired by the Learning Directed Operating System (LDOS) NSF Expedition summer school.

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

## LLM USAGE DISCLOSURE

The use of large language models (LLMs) is permitted as a general-purpose assistive tool. In preparing this paper, we used an LLM (OpenAI's ChatGPT) to assist with polishing the exposition, including improving grammar, conciseness, and style in several sections of the manuscript. The LLM was not used for research ideation, technical contributions, or generation of novel results. All conceptual development, proofs, and experiments were conducted by the authors.

## ETHICS STATEMENT

This work studies dynamic indexing strategies for cascaded inference. It does not involve human subjects, sensitive personal data, or proprietary datasets, and therefore raises no direct ethical concerns. We believe the contributions align with the ICLR Code of Ethics, and we are not aware of any issues regarding fairness, privacy, security, or potential misuse beyond those generally applicable to large-scale machine learning research.

APPENDIX

## A    MORE DETAILS FROM LITERATURE REVIEW

**Early Exit Models**. Among intra-model cascaded inference methods, *early-exit* (EE) architectures have been widely adopted to accelerate inference in both computer vision and natural language processing, including ResNet (He et al., 2016), VGG (Simonyan & Zisserman, 2015), and BERT-based (Devlin et al., 2019b) models (see Rahmath P et al. (2024); Laskaridis et al. (2021) for a recent survey). The ramp architectures in these models are typically designed to align with the structural properties of the backbone. Existing exit strategies commonly rely on metrics such as label confidence (Xin et al., 2021), prediction entropy (Xin et al., 2020; Teerapittayanon et al., 2016), or more advanced mechanisms such as ramp-level counters (Zhou et al., 2020). However, these approaches do not allow for *prediction with recall*, where the system may return to the output of a previously visited exit—a setting where empirical evidence suggests that earlier exits can occasionally outperform later ones (Kaya et al., 2019).

**Relation to Pandora's Box and Prophet Inequality**. The Pandora's Box framework (Weitzman, 1979; Boodaghians et al., 2020; Chawla et al., 2020) and the Prophet Inequality framework (Krengel & Sucheston, 1977; 1978; Livanos & Mehta, 2024) are two classical models for decision making under uncertainty. Specifically, our exit-with-recall model bears structural similarity to the Pandora's Box problem, while the no-recall variant aligns closely with the Prophet Inequality setting—both situated within a cost minimization framework. However, our setting imposes two key constraints: (i) a precedence constraint on the order in which boxes (or exits) may be inspected, and (ii) a Markovian correlation that the underlying distributions conform to. These constraints render existing algorithms for the aforementioned problems inapplicable.

Since Weitzman's classic formulation (Weitzman, 1979), Pandora's Box has been extended in numerous ways. Variants with order constraints (Boodaghians et al., 2020) and correlated rewards (Chawla et al., 2019; 2020; 2021; Gergatsouli & Tzamos, 2023) highlight the challenges of adaptivity, with the latter proving constant-factor approximation NP-hard in the fully adaptive case. Online variants connect to bandits and set cover, yielding competitive or regret-minimizing algorithms (Gergatsouli & Tzamos, 2022; Gatmiry et al., 2024). Nonobligatory inspection (Doval, 2018; Beyhaghi & Kleinberg, 2019; Beyhaghi & Cai, 2023; Fu et al., 2023) further departs from Weitzman's ranking-based solution, with NP-hardness results and PTAS guarantees. Other extensions address partial openings (Aouad et al., 2020), generalized objectives (Olszewski & Weber, 2015), deadlines (Berger et al., 2024), time-dependent costs (Amanatidis et al., 2024), and strategic information revelation (Ding et al., 2023).

Prophet inequalities, first studied by Krengel & Sucheston (1977; 1978), compare the performance of an online stopping rule to a prophet with full foresight. Classical results (reward maximization) guarantee a $1/2$-approximation in general settings, with improvements under additional structure. Recent work has extended this line to matroids and combinatorial constraints (Kleinberg & Krakovski, 2005; Bhattacharya & Khanna, 2012), correlated and non-i.i.d. distributions (Correa et al., 2019; Yan, 2011), and online matching and allocation problems (Esfandiari et al., 2017; Feldman et al., 2016). Further advances include connections to posted pricing in mechanism design (Chawla et al., 2010; Correa et al., 2022) and to multi-armed bandits (Gatmiry et al., 2024).

**Data-Driven Algorithm Design**. Our framework relates to data-driven algorithm design (Gupta & Roughgarden, 2016) with cost, where practitioners refine parameterized algorithms via training instances to maximize expected future performance. Data-driven algorithm design Gupta & Roughgarden (2016) includes greedy heuristic selection, self-improving algorithms (Clarkson et al., 2010; Ailon et al., 2011), and parameter tuning in optimization and machine learning Goel et al. (2006); Bergstra & Bengio (2012); Snoek et al. (2012); Jamieson & Talwalkar (2016); Balcan et al. (2017); Li et al. (2017); Kleinberg et al. (2017); Hazan et al. (2018); Weisz et al. (2018); Balcan et al. (2018); Alabi et al. (2019); Balcan et al. (2019); Kotthoff et al. (2019); Sivaprasad et al. (2020).

**Gittin's Index** In particular, in the single-line and multi-line cases our adaptive index reduces to the well-known non-*discounted* Gittins index (Gittins, 1979; 1989; Weber, 1992). By contrast, for directed-tree and skip-graph structures, the indexing strategy developed here appears to be novel, as no analogous formulation has been established in prior work. Other recent studies have also

leveraged Gittins indices to address efficiency challenges in machine learning, including applications such as hyperparameter tuning (Xie et al., 2024; 2025).

**Other Related Work**. Another relevant direction is the *stochastic probing* problem, where one must decide both which elements to probe and when to probe them. Gupta & Nagarajan (2013) introduced this setting under matroid and knapsack intersection constraints, giving the first polynomial-time $\Omega(1/k)$-approximate sequential posted price mechanism for $k$-matroid intersections. Adamczyk et al. (2016) extended the model to monotone submodular objectives, achieving a $\frac{1-1/e}{k_{\mathrm{in}}+k_{\mathrm{out}}+1}$-approximation in general and a tighter $\frac{1}{k_{\mathrm{in}}+k_{\mathrm{out}}}$ bound for linear objectives. Subsequent work (Gupta et al., 2016; 2017) investigated adaptivity gaps, quantifying the performance loss of non-adaptive strategies relative to optimal adaptive ones under prefix-closed constraints.

Beyond stochastic probing, other problems share similar information structures or solution concepts, including search (Armstrong, 2017; Kleinberg & Kleinberg, 2018), ranking (Derakhshan et al., 2022), Markov games (Li & Liu, 2022), sorting and selection (Gupta & Kumar, 2001), revenue maximization (Kleinberg et al., 2016; Chawla et al., 2019), and costly information acquisition (Charikar et al., 2000; Chen et al., 2015b;a; Li & Shi, 2017; Singla, 2018; Bergemann et al., 2018; Gupta et al., 2019; Chawla et al., 2024).

## B MORE DETAILS FROM SINGLE LINE COSTLY EXPLORATION

### B.1 MORE DETAILS FOR THE DYNAMIC INDEX

We introduce additional notation to facilitate the formal analysis of the dynamic index. Specifically, we use $\Phi$ and $\phi$ interchangeably to denote the equivalent loss.

**Lemma B.1** (Properties of $\Phi$ and $H_i$). *Given any state $(x, R_{i-1}, i)$,*

- *$\Phi(\cdot, R_{i-1}, i)$ is 1-Lipschitz and monotone non-decreasing.*
- *Let $H_i(x, R_{i-1}) := \Phi(x, R_{i-1}, i) - x$, then $H_i(\cdot, R_{i-1})$ is nonnegative, 1-Lipschitz and monotone non-increasing.*
- *For $\sigma_i$ as the index (Def. 4.4) of the $i$-th node of the hypernode, then $\Phi(x, R_{i-1}, i) = x$ for any $x \geq \sigma_i$.*

*Proof.* Given any $a < b$,

$$
\Phi(b, R_{i-1}, i) - \Phi(a, R_{i-1}, i)
$$
$$
\leq \mathbb{E}\left[\min\{b, \min_{j=i}^{\tau^*(a, R_{i-1}, i)} R_j\} - \min\{a, \min_{j=i}^{\tau^*(a, R_{i-1}, i)} R_j\}\right]
$$
$$
\leq b - a
$$

where in the first inequality, we used that $\tau^*(a, R_{i-1}, i)$ is a suboptimal strategy for $\Phi^\tau(b, R_{i-1}, i)$. Using the same reasoning, we have:

$$
\Phi(b, R_{i-1}, i) = \mathbb{E}\left[\min\{b, \min_{j=i}^{\tau^*(b, R_{i-1}, i)} R_j\}\right] + \sum_{j=i}^{\tau^*(b, R_{i-1}, i)} c_j
$$
$$
\geq \mathbb{E}\left[\min\{a, \min_{j=i}^{\tau^*(b, R_{i-1}, i)} R_j\}\right] + \sum_{j=i}^{\tau^*(b, R_{i-1}, i)} c_j
$$
$$
\geq \mathbb{E}\left[\min\{a, \min_{j=i}^{\tau^*(a, R_{i-1}, i)} R_j\} + \sum_{j=i}^{\tau^*(a, R_{i-1}, i)} c_j\right] = \Phi(a, R_{i-1}, i)
$$

Thus, $\Phi(\cdot, R_{i-1}, i)$ is monotone non-decreasing. Now consider $H_i$, we have

$$
H_i(b, R_{i-1}) - H_i(a, R_{i-1}) = \Phi(b, R_{i-1}, i) - \Phi(a, R_{i-1}, i) - (b - a) \leq 0
$$

where we use that $\Phi(\cdot, R_{i-1}, i)$ is 1-Lipschitz. The above inequality implies that $H_i(x, R_{i-1})$ is 1-Lipschitz and monotone non-increasing. Lastly, $\Phi(x, R_{i-1}, i) - x = 0$ for all $x \geq \sigma_i$ follows from the that fact that $\sigma_i$ is the smallest such that $H_i(\sigma_i, R_{i-1}) = 0$ and $H_i$ is non-negative and monotone non-increasing.

$\square$

**Lemma B.2** (Properties of the Dynamic Index). *Given a costly exploration problem with line precedence graph $\mathcal{L} = [b_1, \ldots, b_n]$, the dynamic index of every node $i \in [n]$ satisfies the following property: Given any state $R_{i-1}$ as the state of $(i-1)$-th node,*

- *$\sigma_i(R_{i-1}, i)$ is nonincreasing as additional nodes are appended to $\mathcal{L}$.*

- *Let $\eta$ be the (random) index of the first node that has dynamic index larger than $\sigma_i(R_{i-1}, i)$, then $\sigma_i(R_{i-1}, i)$ depends only on the (sub)hypernode $\widehat{\mathcal{L}} := \{b_i, \ldots, b_\eta\}$. If $i = \eta$ with probability 1, then $\sigma_i(R_{i-1}, i)$ depends only on $b_i$.*

*Proof.* • The first property holds because the optimal policy stops at the additional nodes only if they yield a lower expected loss. Consequently, appending nodes at the end of $\mathcal{L}$ can only decrease the expected loss for any given state. Consequently, this operation leads to a nonincreasing dynamic index.

- The second property is due to that the optimal stopping time will stop at $(\eta - 1)$-th node, hence the dynamic index doesn't depend on any nodes starting from $\eta$.

$\square$

**Lemma B.3.** *The smallest solution to (1) exists, and hence Definition 4.4 is well defined. Given current state $(x, R_{i-1}, i)$, if the dynamic index $\sigma_i = x$, then there exists some optimal stopping time $\tau^*(x, R_{i-1}, i) \geq i$.*

*Proof.* Given any state $(x, R_{i-1}, i)$, consider function

$$H_i(x, R_{i-1}) = \Phi(x, R_{i-1}, i) - x$$

$H_i(x)$ is 1-Lipschitz and monotone non-increasing by lemma B.1. Since $H_i(0, R_{i-1}) = \Phi(0, R_{i-1}, i) \geq 0$ and $H_i(v_k, R_{i-1}) = 0$, there exist some $\sigma_i \in S$, such that $H_i(\sigma_i, R_{i-1}) = 0$. This proves the existence of $\sigma_i$.

Now, we show that if $x = \sigma_i$ is positive, then there exists an optimal stopping rule that proceeds to open $b_i$. Fix any $i$ such that $\sigma_i > 0$. Let $\widetilde{\tau}$ be the best strategy among all strategies that open $b_i$. To show that $\widetilde{\tau}$ is indeed optimal, we show that

$$\delta = \Phi(\sigma_i, R_{i-1}, i) - \Phi^{\widetilde{\tau}}(\sigma_i, R_{i-1}, i) = 0$$

Assume towards contradiction that $\delta > 0$. We have

$$\begin{aligned}
0 < \delta &= \Phi(\sigma_i, R_{i-1}, i) - \Phi^{\widetilde{\tau}}(\sigma_i, R_{i-1}, i) \\
&\leq \Phi(\sigma_i, R_{i-1}, i) - \Phi^{\tau^*(\sigma_i - \epsilon, R_{i-1}, i)}(\sigma_i, R_{i-1}, i) \\
&= (\Phi(\sigma_i, R_{i-1}, i) - \Phi(\sigma_i - \epsilon, R_{i-1}, i)) + (\Phi(\sigma_i - \epsilon, R_{i-1}, i) - \Phi^{\tau^*(\sigma_i - \epsilon, R_{i-1}, i)}(\sigma_i, R_{i-1}, i)) \\
&\leq 2\epsilon
\end{aligned}$$

where we used Lipschitzness of $\Phi$ for the last inequality, and the first inequality comes from the fact that $\tau^*(\sigma_i - \epsilon, R_{i-1}, i)$ is a sub-optimal policy that opens $b_i$. We have $\tau^*(\sigma_i - \epsilon, R_{i-1}, i) \geq i$ since $\sigma_i$ is the smallest such that $H_i(\sigma_i, R_{i-1}) = 0$, this implies that $H_i(\sigma_i - \epsilon, R_{i-1}) = \Phi^{\tau^*(\sigma_i - \epsilon, R_{i-1}, i)}(\sigma_i - \epsilon, R_{i-1}, i) - (\sigma_i - \epsilon) > 0$ meaning the optimal policy will accumulate more reward than current best, thus it has to open $b_i$. As $\epsilon \to 0$, we get a contradiction. On the other hand, $H_i(\sigma_i, R_{i-1}) = \Phi(\sigma_i, R_{i-1}, i) - \sigma_i = 0$ implies that the strategy that stops at $b_{i-1}$ is also optimal. Thus, $\sigma_i$ is indeed the value for which we are indifferent between stopping and proceeding optimally. $\square$

### B.2 PAYOFF TABLE

**Lemma B.4** (Efficient Computation of Payoff Table). *There is an efficient algorithm that computes $\phi(x, s, i)$ for all $i$, $x$ and $s$.*

*Proof.* Now we give an efficient algorithm for computing the dynamic index. In fact, we will give an algorithm that uses dynamic programming to compute $\Phi(x, R_{i-1}, i)$ for all triples $(x, R_{i-1}, i)$. Then, given the current state of the algorithm $(x, R_{i-1}, i)$, the dynamic index $\sigma_i$ for node $i$ is the smallest $x$ in the table where $\Phi(x, R_{i-1}, i) = x$.

Denote by $T(x, R_{i-1}, i)$ our three dimensional dynamic programming table. Each entry $T(x, R_{i-1}, i)$ will store the following information:

1. Expected future loss: $\Phi(x, R_{i-1}, i)$

2. Indicator: $\mathbb{1}(x, R_{i-1}, i)$ indicating whether the optimal policy will open $b_i$ in this state

3. The distribution of future random min reward[3]: $R_{\text{FRM}}(x, R_{i-1}, i) := \min_{j=i}^{\tau^*(x, R_{i-1}, i)} R_j$ where $R_j$'s are the correlated random rewards for mininodes that are yet to be opened given that the algorithm is at state $(x, R_{i-1}, i)$.

---

[3] The randomness comes from both random stopping time $\tau^*$ and correlated random variables $R_i$'s,

---

**Algorithm 2** Expected Equivalent Reward Computation, Single Line

---

**Require:** Ordered set of nodes $\{b_1, \ldots, b_n\}$, probing cost $\{c_1, \ldots, c_n\}$, distributions of the random payoff of nodes
1: Initialize $z \leftarrow 0$
2: **for** $x \in S$ **do**                                    ▷ Base case: filling in $T(\cdot, \cdot, n)$
3:     **for** $s \in S$ **do**
4:         $z \leftarrow \sum_{y \in S}(\min\{x, y\} + c_n) \cdot \Pr(R_n = y)$
5:         **if** $z > x$ **then**
6:             $\Phi(x, s, n) = z, \mathbb{1}(x, s, n) = 1$
7:         **else**
8:             $\Phi(x, s, n) = x, \mathbb{1}(x, s, n) = 0$
9:         **end if**
10:         $R_{\text{FRM}}(x, s, n) = \mathbb{1}(x, s, n) \cdot R_n$, and $c_{\text{FR}}(x, s, n) = \mathbb{1}(x, s, n) \cdot c_n$
11:     **end for**
12: **end for**
13: **for** $i = n-1, \cdots, 1$ **do**                          ▷ Filling in $T(\cdot, \cdot, i)$ for all $i = n-1, \cdots, 1$
14:     **for** $x \in S$ **do**
15:         **for** $s \in S$ **do**
16:             $z \leftarrow \mathbb{E}\left[\sum_{y \in S}\left(\min\left\{x, y, R_{\text{FRM}}(x, s_y, i+1)\right\} + c_j + c_{\text{FR}}(x, s_y, i+1)\right) \cdot \Pr(R_i = y)\right]$ where $s_y$ is the state that gives $R_i$ realization $R_i = y$
17:             **if** $z > x$ **then**
18:                 $\Phi(x, s, i) = z, \mathbb{1}(x, s, i) = 1$
19:             **else**
20:                 $\Phi(x, s, i) = x, \mathbb{1}(x, s, i) = 0$
21:             **end if**
22:             Calculate $R_{\text{FRM}}(x, s, i)$ and $c_{\text{FR}}(x, s, i)$ as follows: with probability $\Pr(R_i = y)$, $R_{\text{FRM}}(x, s, i)$ is $\mathbb{1}(x, s, i) \cdot \min\{y, R_{\text{FRM}}(x, s_y, i+1)\}$ and $c_{\text{FR}}(x, s, i)$ is $\mathbb{1}(x, s, i) \cdot (c_i + c_{\text{FR}}(x, s_y, i+1))$
23:         **end for**
24:     **end for**
25: **end for**
26: **Return** $\Phi(x, s, i)$ for all $x \in S$, $s \in S$ and $i \in [n]$

---

4. The distribution of future random cost[4]: $c_{\text{FR}}(x, R_{i-1}, i) := \sum_{j=i}^{\tau^*(x, R_{i-1}, i)} c_j$

Algorithm 2 describes how to fill in the dynamic programming table. Since all random variables that appear in Algorithm 2 has finite support with size bounded by $\text{poly}(K, n)$, and any min operation for random variables only has three or less arguments, it follows that algorithm 2 takes polynomial time and space.                                                                    □

---

[4]The randomness comes from $\tau^*$ being a random stopping time.

## C    MORE DETAILS COSTLY EXPLORATION OVER TREE

**Notations** In our analysis, $\lambda c_i$ always appears as a whole. Notice that in our analysis, $\lambda$ is not used as the tradeoff parameter but is applied for other purposes throughout our analysis on the indexing policy. For graphs with a directed-tree constraint, since each node has exactly one parent, we can allocate the edge cost to the node itself. In this way, inspecting a node $i$ reveals both its loss $\ell_i$ and its inspection cost $c_i$. Please see Fig. 7 as an example.

We first define the notion of a hypernode, namely a directed-line subgraph of the DAG that, under suitable conditions, can be reparameterized as a single node with aggregated loss and cost.

**Definition C.1** (Hypernode). *Given an instance of the costly exploration problem with $n$ nodes $\mathcal{V}$ organized in a directed acyclic graph (DAG) $G = (\mathcal{V}, E)$, a hypernode is a subset of nodes $\mathcal{H} := \{v_1, \ldots, v_m\} \subseteq \mathcal{V}$ such that the subgraph of $G$ induced by $\mathcal{H}$, denoted by $\mathcal{L}$, forms a directed path. That is, for each $i \in [m-1]$, the edge $(v_i, v_{i+1})$ belongs to $E$.*

### C.1    COSTLY EXPLORATION OVER MULTI LINES

Before presenting our solution for the tree case, we first introduce the solution for the multi-line setting. We begin by introducing the problem definition of the multi-line setting.

**Definition C.2** (Multi-Line Costly Exploration). *Given an instance of the Markovian costly exploration problem (Prob. 2.4), we say it is in a multi-line setting if the induced subgraph of $G$, excluding the dummy root node $v_0$, can be decomposed into a collection of disjoint directed lines:*

$$\mathcal{L} = \{L_1, L_2, \ldots, L_k\}, \quad L_j = (v_1^j \to v_2^j \to \cdots \to v_{m_j}^j),$$

*such that every node $v \in \mathcal{V}$ belongs to exactly one line $L_j$. Each line $L_j$ can then be reparameterized as a sequence of hypernodes (Def. C.1), so that the costly exploration problem reduces to choosing among multiple directed lines.*

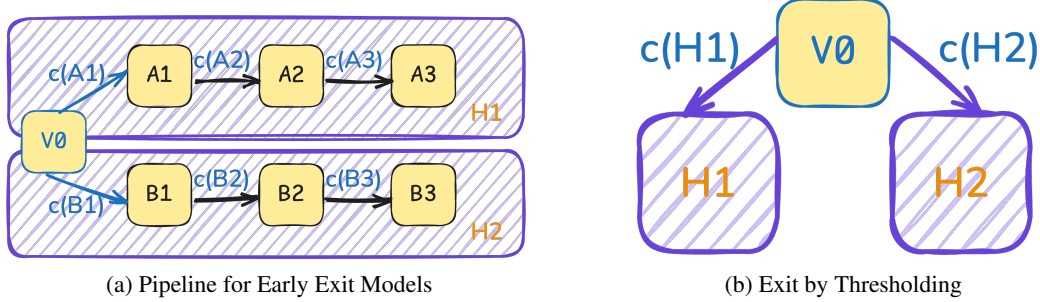

(a) Pipeline for Early Exit Models                    (b) Exit by Thresholding

Figure 7: Hypernode View of multi-line setting. (a) illustrates how to allocate the edge cost $c$ as the inspection cost of the node (b) depicts they way we treat . During inference, the decision rule ($\Diamond$) is the only component that can be modified.

We now show that in the multi-line case, the optimal strategy is to probe each hypernode according to the dynamic index of its first unopened node.

We begin by presenting the definition of a node with random cost, with reward and cost correlated.

**Definition C.3** (Node with Random Cost). *A node $v$ is classified as a node with random cost if it is associated with a loss $\ell$ and an inspection cost $c$, where both $\ell$ and $c$ are random variables drawn from known distributions $\mathcal{D}_\ell$ and $\mathcal{D}_c$, respectively. The inspection cost $c$ may vary and can be correlated with the loss $\ell$; we denote their joint distribution as $\Gamma$.*

Next, we introduce how to equivalently represent a hypernode as a single node with random cost, where the dynamic index of the hypernode equals that of the equivalent node.

**Lemma C.4** (Equivalent Single Node for Hypernode). *For a stopping time $\tau$ and a hypernode $\mathcal{H} := \{v_1, \ldots, v_n\}$, there exists a node $\widehat{v}$ with random cost (Def. C.3) such that following $\tau$ over $\mathcal{H}$ yields the same loss distribution as inspecting $\widehat{v}$.*

*Proof.* Let $L = (\ell_1, \ldots, \ell_n)$ be a realization of the joint loss distribution in hypernode $\mathcal{H}$. For each realization of $L$, the stopping time $\tau$ uniquely determines the effective loss and cumulative cost of the hypernode.

To construct the distribution of the equivalent single node, we define a coupling between the realizations of $L$ and the resulting loss and cost. When the joint loss is $L$, we assign the single node's loss as $\min_{i=1}^{\tau} \ell_i$ and the single node's cost as $\sum_{i=1}^{\tau} c_i$, where $\tau$ is the stopping time. The probability of each outcome matches the probability of $L$ under the original hypernode's joint distribution. $\qquad\square$

From our construction of the equivalent node, we obtain the following lemma: the dynamic index remains well-defined for a node with random cost, even when the nodes inside a hypernode have stochastic costs.

**Lemma C.5** (Extending Dynamic Index to Hypernodes with Random Cost). *Given a hypernode $\mathcal{H} := \{v_1, \ldots, v_n\}$, where each node has a stochastic inspection cost that may be correlated with its loss distribution, the dynamic index of each node with random cost is well-defined and can be computed in polynomial time.*

*Moreover, if the dynamic index $\widehat{\sigma}$ of individual nodes—i.e., the index when there is only one node with random cost—satisfies*

$$\widehat{\sigma}(v_1) \le \widehat{\sigma}(v_2 \mid \ell_1) \le \cdots \le \widehat{\sigma}(v_n \mid \ell_{n-1}),$$

*for any realized losses $\ell_1, \ldots, \ell_n$, then the dynamic index of $v_i$ within the hypernode depends only on $v_i$ itself.*

*Proof.* Since the $\phi$ and $H$ functions remain well-defined and preserve their properties, the dynamic index remains well-defined for hypernodes with nodes of random cost. The second claim follows from Lem. B.2. $\qquad\square$

We begin by presenting a key lemma for our main theorem, which establishes that under certain Markovian correlations, the dynamic index remains an optimal decision rule. We show this lemma by first principles, where we compare the utility of the ordering $A \prec B \prec C$ with that of $B \prec A \prec C$ through a case-by-case analysis of 9 outcomes from the joint distribution of $A$, $B$, and $C$ and aggregate them by the law of total expectation.

**Lemma C.6** (Probing Equivalent Nodes). *Consider three nodes $A, B, C$ with random cost (Def. C.3) satisfying the following properties:*

- *The loss and cost of $B$ are independent of the loss and cost of $A$;*

- *The loss and cost (hence effective outcome) of $C$ depend on both $A$ and $B$ in a Markovian fashion;*

- *The dynamic indices satisfy $\sigma(A) < \sigma(B) < \sigma(C)$ given any realizations of $A$ and $B$[5], i.e., for any possible value $x$ of $A$ and value $y$ of $B$,*

$$\sigma(A) \le \sigma(B) \le [\sigma(C) \mid \ell_A = x, \ell_B = y];$$

- *A precedence constraint that $A$ and $B$ must be probed before $C$;*

*then, conditioned on any competing loss $X$, the optimal probing strategy is $A \prec B \prec C$.*

*Proof.* It's sufficient to compare two strategies: 1) $D_1 : B \to A \to C$, and 2) $D_2 : A \to B \to C$. Notice that if $X < \sigma(A)$, then it's optimal to not probe any node, then the ordering of the node doesn't matter. WLOG, we may assume $X > \sigma(A)$.

We first write down the expected loss according to ordering strategy $D_1$, if we use notation as in Table. 1, we have that this loss is equivalent to:

$\mathbb{E}[c_B] + \mathbb{E}[R_B|\pi_B]\Pr[\pi_B]$
$+ \lambda_B[\mathbb{E}[c_A] + \mathbb{E}[R_A|\pi_A]\pi_A + \mathbb{E}[\min\{R_A, R_B, y\}|\lambda_B \cap \lambda_A]\lambda_A] + \mathbb{E}[\min\{R_B, y|\rho_A, \lambda_B\}\rho_A]]$
$+ \rho_B[\mathbb{E}[c_A] + \mathbb{E}[R_A|\pi_A]\pi_A + \mathbb{E}[\min\{y, R_A|\lambda_A\}\lambda_A + \mathbb{E}[\phi_C(\{R_A, R_B, X\}|\rho_A \cap \rho_B)]]]$

---

[5]Here, $\sigma(C)$ is a random variable that depends on $A$ and $B$.

Here, we abuse the notation $\rho, \pi, \lambda$ to denote both events and their probabilities, with the intended meaning clear from context; we use $\phi$ as similar definition to $\Phi$ (Lem. B.1), which denote the equivalent loss of probing $C$ based on the observation. We also use $E \cap F$ to denote the event that both $E$ and $F$ occur.

Similarly, using the notations in Table 2, we have that the expected loss according to strategy $D_2$ is:

$$E[c_A] + \pi_A \mathbb{E}[R_A|\pi_A] + \lambda_A \mathbb{E}[\min\{X, R_A\}|\lambda_A]$$
$$+ \rho_A[\mathbb{E}[c_B] + \pi_B \mathbb{E}[R_B|\pi_B] + \lambda_B \mathbb{E}[\min\{X, R_B\}|\lambda_B \cap \rho_A] + \rho_B \phi_C(\min\{X, R_B, R_A\}|\rho_A \cap \rho_B)]$$

Notice that the last term of both payoffs can be cancelled out. Also notice that the reservation value for box $A$ and $B$ satisfies:

$$\mathbb{E}[(\sigma_B - R_B)_+ - c_B] = 0$$

Plugging in the appropraite values of $\rho, \pi, \lambda$, we have:

$$\mathbb{E}[c_A] = \pi_A \sigma_A - \pi_A \mathbb{E}[R_A|\pi_A]$$
$$\mathbb{E}[c_B] = (\pi_B + \lambda_B)\sigma_B - \pi_B \mathbb{E}[R_B|\pi_B] - \lambda_B \mathbb{E}[R_B|\lambda_B]$$

Now, plugging the value of the expected cost and after simplification, we have that:

$$\mathbb{E}[\text{UTIL}(D_2) - \text{UTIL}(D_1)]$$
$$= \pi_B \pi_A (\sigma_A - \sigma_B)$$
$$+ \pi_A \lambda_B [\mathbb{E}[R_B|\lambda_B] - \sigma_B] + \lambda_A \pi_B [\mathbb{E}[\min\{X, R_A|\lambda_A\} - \sigma_B]]]$$
$$+ \lambda_B \lambda_A [-\mathbb{E}[\min\{R_A, R_B.X\}|\lambda_B \cap \lambda_A] - \sigma_B$$
$$+ \mathbb{E}[R_B|\lambda_B] + \mathbb{E}[\min\{y, R_A\}|\lambda_A]]$$
$$< \lambda_B \lambda_A [-\mathbb{E}[\min\{R_A, R_B.X\}|\lambda_B \cap \lambda_A] - \sigma_B + \mathbb{E}[R_B|\lambda_B] + \mathbb{E}[\min\{y, R_A\}|\lambda_A]]$$

where the last inequality follows by the property that $\mathbb{E}[A|A \leq X] < X$.

Finally, we show that the last term is negative. Notice that:

$$\mathbb{E}[\min\{R_A, R_B, X\}|\lambda_B \cap \lambda_A]$$
$$= \sigma_B + \mathbb{E}[\min\{\min\{R_A, X\} - \sigma_B, R_B - \sigma_B\}|\lambda_B \cap \lambda_A]$$
$$\leq \sigma_B + \mathbb{E}[\min\{R_A, X\} - \sigma_B + R_B - \sigma_B|\lambda_B \cap \lambda_A]$$
$$= \mathbb{E}[R_B|\lambda_B] + \mathbb{E}[\min\{X, R_A\}|\lambda_A] - \sigma_B < 0$$

where the last equality follows from the independence of box $A$ and $B$. Aggregating all of the above we have:

$$\mathbb{E}[\text{UTIL}(D_2) - \text{UTIL}(D_1)] < 0.$$

Hence it's optimal to probe according to policy $D_2$. $\qquad\square$

| | $R_A \leq \sigma_A$ $\pi_A$ | $R_A \in (\sigma_A, \sigma_B)$ $\lambda_A$ | $R_A \geq \sigma_B$ $\rho_A$ |
|---|---|---|---|
| $R_B \leq \sigma_A$ $\pi_B$, stop at $B$. | $\mathbb{E}[R_B|\pi_B]$ $+ \mathbb{E}[c_B|\pi_B]$ | $\mathbb{E}[R_B|\pi_B]$ $+ \mathbb{E}[c_B|\pi_B]$ | $\mathbb{E}[R_B|\pi_B]$ $+ \mathbb{E}[c_B|\pi_B]$ |
| $R_B \in (\sigma_A, \sigma_B)$ $\lambda_B$, open $A$. | $\mathbb{E}[R_A|\pi_A]$ $+ \mathbb{E}[c_B|\lambda_B] + \mathbb{E}[c_A|\pi_A]$ | $\mathbb{E}[\min\{R_A, R_B, X|\lambda_A \cap \lambda_B\}]$ $+ \mathbb{E}[c_B|\lambda_B] + \mathbb{E}[c_A|\lambda_A]$ | $\mathbb{E}[\min\{R_B, y\}|\lambda_B]$ $+ \mathbb{E}[c_B|\lambda_B] + \mathbb{E}[c_A|\rho_A]$ |
| $R_B \geq \rho_B$ $\rho_B$ | $\mathbb{E}[R_A|\rho_B]$ $+ \mathbb{E}[c_B|\rho_B] + \mathbb{E}[c_A|\pi_A]$ | $\mathbb{E}[\min\{X, R_A\}|\lambda_A]$ $+ \mathbb{E}[c_B|\rho_B] + \mathbb{E}[c_A|\lambda_A]$ | $\mathbb{E}[\phi_C(\{R_A, R_B, X\}|\rho_A \cap \rho_B)]$ $+ \mathbb{E}[c_B|\rho_B] + \mathbb{E}[c_A|\rho_A]$ |

Table 1: Table for Expected Payoff according to Strategy $D_1$. This table presents the expected total payoff for every possible joint distribution of box $A$ and $B$. In addition, $\rho_A = 1 - \pi_A - \lambda_A$.

|  | $R_A \leq \sigma_A$
$\pi_A$, stop at $A$ | $R_A \in (\sigma_A, \sigma_B)$
$\lambda_A$, stop at $A$ | $R_A \geq \sigma_B$
$\rho_A$ |
|---|---|---|---|
| $R_B \leq \sigma_A$
$\pi_B$ | $\mathbb{E}[R_A|\pi_A]$
$+\mathbb{E}[c_A|\pi_A]$ | $\mathbb{E}[\min\{X,R_A\}|\lambda_A]$
$+\mathbb{E}[c_A|\lambda_A]$ | $\mathbb{E}[R_B|\pi_B]$
$+\mathbb{E}[c_A|\rho_A]+\mathbb{E}[c_B|\pi_B]$ |
| $R_B \in (\sigma_A, \sigma_B)$
$\lambda_B$ | $\mathbb{E}[R_A|\pi_A]$
$+\mathbb{E}[c_A|\pi_A]$ | $\mathbb{E}[\min\{X,R_A\}|\lambda_A]$
$+\mathbb{E}[c_A|\lambda_A]$ | $\mathbb{E}[\min\{X,R_B\}|\lambda_B \cap \rho_A]$
$+\mathbb{E}[c_B|\lambda_B]+\mathbb{E}[c_A|\rho_A]$ |
| $R_B \geq \rho_B$
$\rho_B$ | $\mathbb{E}[R_A|\pi_A]$
$+\mathbb{E}[c_A|\pi_A]$ | $\mathbb{E}[\min\{X,R_A\}|\lambda_A]$
$+\mathbb{E}[c_A|\lambda_A]$ | $\phi_C(\min\{y,R_B,R_A\}|\rho_A \cap \rho_B)$
$+\mathbb{E}[c_B|\rho_B]+\mathbb{E}[c_A|\rho_A]$ |

Table 2: Table for Expected Payoff according to Strategy $D_2$. This table presents the expected total payoff for every possible joint distribution of box $A$ and $B$. In addition, $\rho_B = 1 - \pi_B - \lambda_B$.

**Proof Of Corretness and Time Complexity** We now establish the optimality of the dynamic index in the multi-line setting, showing that probing nodes in the order of their current dynamic indices maximizes the expected utility.

**Theorem C.7** (Dynamic Index for Multi-Line Costly Exploration). *Given a Markovian multi-line costly exploration instance with $m$ lines, the optimal strategy for minimizing expected loss is to probe the uninspected but available nodes with the minimum dynamic index.*

*Proof.* Let $n^*$ denote the node with the smallest dynamic index before any nodes are probed, and let $\sigma_{n^*}^0$ be the value of this index. Let $n \neq n^*$ denote another node. Without loss of generality, we treat the dynamic index of a node as the index of its first unopened child, conditioned on realized losses. Let $\pi^*$ denote the policy that always probes the node with the smallest current dynamic index.

We prove that no strategy outperforms $\pi^*$ by induction on the number of nodes.

**Base Case.** For two nodes, it is immediate that $\pi^*$ is optimal.

**Induction Step.** Assume $\pi^*$ is optimal for $q-1$ nodes; we show it remains optimal for $q$ nodes. If the optimal strategy begins by probing $n^*$, then by induction $\pi^*$ is already optimal.

Suppose instead that the best alternative strategy $\widehat{\pi}$ begins with node $n$ whose index $\sigma_n^0$ is larger than $\sigma_{n^*}^0$. By the induction hypothesis, after probing $n$, strategy $\widehat{\pi}$ must follow $\pi^*$. Thus $\widehat{\pi}$ proceeds as: probe $n$ until its index exceeds $\sigma_{n^*}^0$, then switch to $n^*$, and continue optimally.

Now construct a new strategy $\bar{\pi}$ that starts with $n^*$, switches to $n$ once its index exceeds $\sigma_{n^*}^0$, and then continues as in $\widehat{\pi}$. Both $\widehat{\pi}$ and $\bar{\pi}$ explore the same nodes, but in different orders. Since $\sigma_{n^*}^0 < \sigma_n^0$, Lemma C.6 implies that $\bar{\pi}$ achieves strictly less expected loss than $\widehat{\pi}$, contradicting the optimality of $\widehat{\pi}$.

Thus, by induction, $\pi^*$ is optimal for any number of nodes. $\square$

Finally, we note that the dynamic index can be implemented in polynomial time and space.

**Theorem C.8** (Polynomial-Time Implementation of Dynamic Indexing, Multi-Line). *The dynamic index for the multi-line setting can be implemented in polynomial time and space. Furthermore, preprocessing this policy takes $O(n \cdot |V|^2 T)$ time and requires $O(n|V|^2)$ space. At inference time, for each input $x$, the policy runs in $O(1)$ per node and $O(n)$ overall per input.*

*Proof.* **Implementation**. One can still use the idea of the dynamic index lookup table as in the single-line setting, since the loss function $\ell$ across lines is independent. Note that the optimal strategy, characterized by the dynamic index, visits at most a finite number of nodes, since the total number of nodes is finite. For each hypernode visit, it suffices to store the dynamic indices of the competing nodes and the index when the strategy last entered this hypernode.

**Complexity**. The space complexity of multi-line case is the same as the single-line setting, in that the space complexity is linear in the number of nodes. Similarly, the time complexity of preprocessing and during inference is the same as the single-line setting. $\square$

## C.2 Extension to Directed Tree

Next, we introduce how to generalize the previous indexing policies from the multi-line setting in order to achieve theoretical optimal performance. Note that the directed tree case is equivalent to the directed forest setting (where each root has only outward edges), since we can add a dummy root node connecting all roots in the forest.

**Graph Basics**. We introduce some basics from graph theory in order to formally define tree contraction.

**Definition C.9** (Component). *Given an undirected graph $G = (V, E)$, a **component** of $G$ is a maximal connected subgraph $C = (V_C, E_C)$ such that:*

- *$C$ is connected: There exists a path between any two vertices in $V_C$.*

- *$C$ is maximal: No additional vertex $v \in V \setminus V_C$ can be included without losing connectivity.*

*A graph is said to be **connected** if it consists of a single component.*

**Definition C.10** (Induced Subgraph). *Given a graph $G = (V, E)$ and a subset of vertices $V' \subseteq V$, the* induced subgraph $G[V']$ *is the graph $(V', E')$ where:*

$$E' = \{(u, v) \in E \mid u, v \in V'\}$$

*That is, $G[V']$ contains all edges from $G$ whose endpoints are both in $V'$.*

We next define a *minimal tree*.

**Definition C.11** (Directed Tree, Branch Vertex, Minimal Tree). *A directed* tree $\mathcal{T}$ *is a connected DAG equipped with an orientation such that there exists a designated root vertex $r$ with the following properties:*

- *every edge is oriented away from dummy root $v_0$ and root $r$.*
- *for every vertex $v \in \mathcal{T}$, there exists a unique directed path from $r$ to $v$.*

*A* branch vertex *in a directed tree is a vertex with at least two outgoing edges. A* minimal tree *is a directed tree whose proper subgraphs contain no branch vertices.*

**Generalizing Dynamic Indexing to Directed Tree**. The key intuition behind our algorithm is that, after probing the root $r$ in a minimal tree, the remaining costly exploration problem reduces to exploring *multiple lines*, enabling us to apply our solutions in Section C.1. We formally define the dynamic index for the directed tree setting.

**Definition C.12** (Dynamic Index, Directed Tree). *Consider a Markovian costly exploration problem with precedence graph $G = (\mathcal{V}, E)$ structured as a directed tree. Let $\mathcal{V}_o$ denote the set of opened nodes, and let the information set on the realized losses be $\mathcal{I}_o := \{v_i = \ell_i \mid i \in \mathcal{V}_o\}$.*

*For any unopened node $v_i$, its dynamic index is derived by applying Alg. 3 on the component of the induced subgraph $G[\mathcal{V} \setminus \mathcal{V}_o]$, conditioned on the current information set $\mathcal{I}_o$.*

*Proof.* When computing the dynamic index for a node inside the algorithm, the node is either the root of a minimal tree, or the current graph reduces to a collection of disjoint lines. It suffices to show that the index is well-defined in both cases.

From the multi-line results, we know that if the graph[6] is a line or a set of isolated nodes, then the dynamic index is well-defined. Moreover, the entire graph can be contracted into a single equivalent node with random cost (Lem. C.4).

We now define the dynamic index of the root $r$ of a minimal tree. First, we contract the induced subgraph consisting of all vertices other than $r$ into a single equivalent node $\widehat{v}$ with random cost. Given the current observed loss $x$, the equivalent loss for $r$ is computed as:

$$\Phi(x, r) = \min \left\{ x, \ \mathbb{E}[\min\{\ell_r, x\}] + c_r, \ \mathbb{E}[\min\{\ell_r, x, \ell_{\widehat{v}}\}] + c_r + c_{\widehat{v}} \right\}.$$

---

[6]In our modeling, the structure of the graph is determined by the induced subgraph obtained after removing the dummy root $r$.

The three terms correspond to: (i) stopping without opening $r$, (ii) opening $r$ only, and (iii) opening $r$ while optimally exploring the remaining nodes.

Consequently, we define the dynamic index of $r$ as the smallest $x$ satisfying $\Phi(x, r) = x$. This index is well defined since the minimal tree can be equivalently treated as a hypernode with $r$ as the first node and $\widehat{v}$ as the second node. By Lem. C.5, the dynamic index is therefore guaranteed to be well defined. $\qquad\square$

In addition, we present our algorithm for computing the dynamic index for every node. The algorithm iteratively identifies and contracts the minimal subtrees of the underlying graph into directed lines, thereby eliminating all branch vertices step by step. Notice that after inspecting a new node, we need to partially update the dynamic index of each of its children. [7] Please see Figure 3 for an illustration.

---

**Algorithm 3** Updating Dynamic Index in Forests

---

**Require:** An instance of a Markovian costly exploration problem with precedence graph $G$ structured as a forest.
1: $\widehat{G} \leftarrow G, t \leftarrow 1$.
2: **while** there exist minimal trees in $\widehat{G}$ **do**
3:     **for** each minimal tree $\mathcal{T}_i$ with root $r_i$ **do**
4:         **for** every possible loss realization $\ell_{r_i}$ of $r_i$ **do**
5:             Condition on $\ell_{r_i}$, compute $\phi$ and the dynamic index $\sigma$ for all states of nodes in $\mathcal{T}_i \setminus \{r_i\}$.
6:             Contract $\mathcal{T}_i \setminus \{r_i\}$ into a single node $\widehat{v}_i$, and compute its loss and cost distribution conditioned on $\ell_{r_i}$.
7:         **end for**
8:         Update $\widehat{G}$ accordingly.
9:     **end for**
10: **end while**
11: Compute the dynamic index for the remaining nodes.

---

**Proof of Correctness and Complexity**.

**Lemma C.13** (Time and Space Complexity of Dynamic Index, Directed Tree). *The dynamic index for the directed tree setting (Algorithm 3) can be implemented in polynomial time and space. Furthermore, preprocessing this policy takes $O(n \cdot |V|^2 T)$ time and requires $O(n|V|^2)$ space. At inference time, for each input $x$, the policy runs in $O(1)$ per node and $O(n)$ overall per input.*

*Proof.* **Preprocessing.** During the execution of the algorithm, each minimal tree can be identified using either a *BFS* or *DFS* traversal, both of which run in $O(n)$. Each tree traversal and possible contraction only needs to be performed once.

Similar to the multi-line case, we maintain an if-stop table based on the minimum loss and the loss of the current box. The main difference is that the computation of the dynamic index for a node now proceeds from the leaf nodes. Consequently, the preprocessing time remains $O(n \cdot |V|^2 T)$, since each leaf has only one parent.

**Space complexity.** We store the $\phi$-table for all possible states of each node, together with the graph structure required by the algorithm. As argued in earlier sections, storing the $\phi$-table requires $O(n \cdot |V|^2)$ space.

**Time complexity.** At inference time, the complexity per input is $O(n)$, since we only need to perform table lookups. $\qquad\square$

Since probing according to the latest dynamic index can probe at most all the nodes, Algorithm 3 is invoked at most once per node in the forest. Thus, the overall runtime is still polynomial.

---

[7]In the presence of already probed nodes, for any unopened node $i$, we apply the algorithm to the subtree rooted at $i$ to compute its dynamic index.

**Theorem C.14** (Optimality of Dynamic Index, Directed Tree). *The dynamic index defined in Def. C.12 is optimal for probing in the forest setting.*

*Proof.* Each contraction step preserves the *equivalent loss table* and the *loss distribution* of probing the minimal tree. Since the algorithm updates $\phi$ for the root of a minimal tree using the strategy that minimizes expected loss, the $\phi$ values—and thus the dynamic indices—of its parent nodes remain unchanged under contraction.

This allows us to reduce the graph $\widehat{G}$ to one consisting only of *multiple lines and isolated nodes*, on which the $\phi$ table and dynamic indices $\sigma$ can be computed directly. Applying Thm. C.7 to $\widehat{G}$ then establishes the optimality of the dynamic index in forests. □

### C.3 COSTLY EXPLORATION WITH SKIP

Next, we introduce the costly exploration setting that allows skipping, which corresponds to the graph structure given by the transitive closure of a directed line. We can still use dynamic programming to pre-compute the if-stop matrix and the equivalent-loss table. The difference is that, when computing the equivalent loss, we must enumerate over all possible next nodes rather than just one. More formally,

$$\Phi(X, R, v_i) = \min\Big\{X, \min_{v \in C(v_i)} c(v) + \mathop{\mathbb{E}}_{R_v | R}[\phi(\min\{X, R_i\}, R_i, v)]\Big\}$$

where $C(v_i)$ denotes the set of nodes that are immediate children of $v_i$. Here, our definition of $\phi$ differs from that in the other sections: its third argument denotes the *current* node, its first argument denotes the minimum loss, and its second argument denotes the current loss.

**Proof of Correctness and Complexity**. Note that the only difference between our algorithm and the single-line case is that the set of candidate nodes to inspect grows from $O(1)$ to $O(n)$, which increases the preprocessing time by a factor of $n$. The inference time, however, remains the same as before, since we still only need to store, for each state, whether to stop at each possible value. Hence, we obtain the following lemma:

**Lemma C.15** (Time and Space Complexity of Dynamic Index, Transitive Closure of Directed Line). *The dynamic index for the transitive closure of a directed line can be implemented in polynomial time and space. In particular, preprocessing this policy takes $O(n^2 \cdot |V|^2 T)$ time and requires $O(n|V|^2)$ space. At inference time, for each input $x$, the policy runs in $O(1)$ per node and $O(n)$ overall per input.*

By Bellman's principle of optimality, the equivalent loss is minimized at every step, from which the following result directly follows.

**Theorem C.16** (Optimality of Dynamic Index, Transitive Closure of Directed Line). *The dynamic index defined in Def. C.12 is optimal for probing in the transitive closure of a directed line.*

## D  MORE DETAILS FROM THE EXPERIMENTS

In this section, we provide additional experimental details omitted from the main text due to space constraints, along with background information on the early exit inference model.

### D.1  PRELIMINARIES ON EARLY EXIT MODELS

Early-exit (EE) models extend standard deep neural networks by introducing multiple intermediate exit points, enabling inference to terminate early on "easy" inputs. An EE model typically consists of three components: (1) a *backbone model*, corresponding to the original single-exit architecture; (2) a set of *ramps*, i.e., intermediate exits attached to selected layers of the backbone; and (3) an *exit decision policy*, which determines whether to stop at a given ramp and return its prediction. Importantly, only the exit policy remains controllable at inference time.

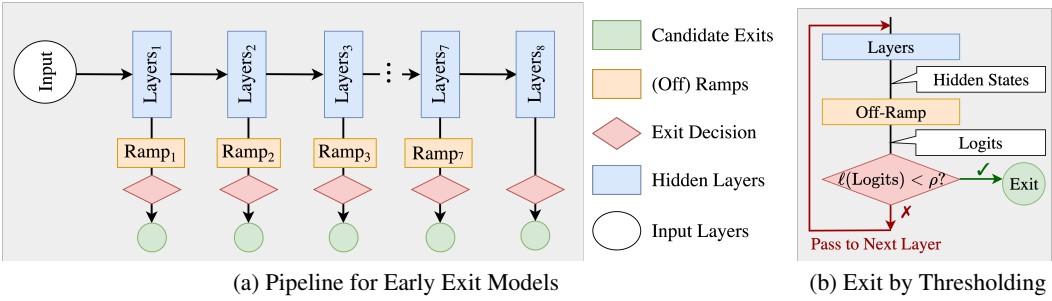

(a) Pipeline for Early Exit Models        (b) Exit by Thresholding

Figure 8: Exit decisions in early-exit models. (a) shows the pipeline of an early-exit model; (b) illustrates the exit decision process under a thresholding strategy. During inference, the decision rule ($\lozenge$) is the only component that can be modified.

**Thresholding.** A common exit policy is thresholding. Each ramp is associated with a fixed threshold. During inference, if the task-specific loss (or, equivalently, if the prediction confidence from the softmax output) at a ramp surpasses the threshold, inference halts and the prediction is returned. See Figure 8(b) for an illustration under classification tasks.

*Confidence and entropy based thresholding strategies arise as natural variants of "no-recall" policies*, under the directed line T-TAMER setting. T-TAMER maintains a stop/no-stop table indexed by the loss at each submodel. If the table assigns "no-stop" to loss values above a given loss threshold, we recover the usual confidence-based rule by setting loss $= 1 -$ confidence. Similarly, we recover the entropy based rule by setting loss $=$ entropy, for nonnegative entropy functions.

**Markovian dependency.** EE models are typically designed to place ramps at "natural breakpoints," such as layer groups in CNNs or transformer blocks. This induces a Markovian structure: the output distribution at each ramp depends only on its immediate predecessor, making the sequence of ramps analogous to a single directed line. This observation underlies our theoretical abstraction, where EE corresponds to the *single-line setting* of our indexing framework.

**Notation.** Let $n$ denote the number of exits (with the final exit being the backbone output). For an input $x$, let $\ell_i(x)$ denote the task-specific loss at the $i$-th exit. We abstract each ramp as an *node*, which will serve as the atomic unit in our theoretical model.

To capture the trade-off between accuracy and efficiency for EE models, we employ a latency-aware loss. The FLOP-based cost can be replaced with any hardware-invariant measure of latency, ensuring consistent policies across platforms.

**Definition D.1** (Latency-Aware Loss). *Given an input $x$, the latency-aware loss of using exit $j \leq i$ after evaluating $i$ exits is*

$$\theta_\lambda(j, i)[x] := (1 - \lambda)\,\ell_j(x) \;+\; \lambda \sum_{k=1}^{i} c_k,$$

*where $c_k := \text{FLOP}(m_k) \geq 0$ is a proxy for the computation cost of sub-model $m_k$. The hyperparameter $\lambda \geq 0$ balances accuracy and latency.*

Specifically, $\lambda$ is a tunable parameter, and the latency cost is measured as the ratio FLOPs(node)/FLOPs(backbone). The accuracy component is defined as $1 - \text{confidence}$, where confidence is the softmax probability of the predicted class.

## D.2 Accelerating T-TAMER via Soft State Aggregation

In this section, we provide an overview of how to use soft state aggregation, a common technique in Markov decision processes for accelerating computations on the transition matrix when the state space contains a large number of discrete values.

To begin with, we briefly outline the existing algorithmic implementation and pinpoint its runtime bottleneck. For the implementation, we employ a matrix-based variant of T-TAMER for the single directed-line setting (Algorithm 2), formalized in Algorithm 4. The procedure first discretizes the loss distribution, and estimates the transition matrix over the discretization support $V$ at each ramp. Then, it performs a backward dynamic-programming step that employs matrix operations to compute the equivalent losses table by comparing the losses incurred under continuation versus stopping. This yields the optimal stop/no-stop decision rule at every state of the algorithm, which is subsequently applied to all trajectories to determine their exit points. In this algorithm, the dominant computational cost arises in line 6, which requires multiplying two $|V| \times |V|$ matrices.

---

**Algorithm 4** T-Tamer Learning Phase

---

**Require:** $N$ samples of loss trajectories over $n$ ramps; discretization set $V$ of loss values, costs $\{c_i\}_{i=1}^n$.

1: Discretize all accuracy values into $|V|$ bins.          $\triangleright \Theta(Nn)$
2: Accumulate empirical joint counts over $N$ samples.        $\triangleright \Theta(N)$
3: For each ramp $i$, construct transition matrix $P_i \in \mathbb{R}^{|V| \times |V|}$.    $\triangleright \Theta(Nn + n|V|^2)$
4: Define matrices $X, S \in \mathbb{R}^{|V| \times |V|}$ as follows:

$$X_{x,s} = x, \qquad S_{x,s} = s \quad \text{for all } x, s \in V,$$

compute $\Phi_n := \min(X, S)$ as the element wise minimum of $X$ and $S$.
              $\triangleright \Theta(|V|^2)$ to form both matrices

5: **for** $i = n - 1, \ldots, 1$ **do**
6:   Compute continuation value via matrix multiplication:

$$C_i = P_i \, \Phi_{i+1} + c_{i+1}.$$

            $\triangleright \Theta(|V|^3)$ (Matrix Multiplication)

7:   Compute stopping value:
$$S_i = S.$$

            $\triangleright \Theta(|V|^2)$

8:   Elementwise minimization and policy extraction:

$$\Phi_i = \min(S_i, C_i), \qquad \text{stop}_i = \mathbf{1}[S_i \leq C_i].$$

            $\triangleright \Theta(|V|^2)$

9: **end for**
10: Evaluate policy $\pi$ on all $N$ trajectories to determine exits.    $\triangleright \Theta(Nn)$

---

This algorithm runs efficiently when $|V|$ is small. When our additive discretization parameter $\epsilon$ satisfies $\epsilon < 0.001$, we typically have $|V| \sim 1000$. In this regime, approximating the transition matrix using the product of two low-rank matrices can substantially accelerate the computation.

*Remark.* In the algorithm 4, we estimate the cost of each matrix multiplication (MM) as $O(|V|^3)$. In fact, modern fast matrix multiplication algorithms reduce this runtime to $O(|V|^\omega)$ for some exponent $\omega \in [2, 2.38)$; see Williams (2012) for a general discussion of how the exponent $\omega$ is used in algorithmic runtime analyses.

Next, we present the intuition behind soft state aggregation. In our setting, a *state* corresponds to an index of the transition matrix—equivalently, a discretized loss level in the support $V$. The intuition of state aggregation is to reduce the size of the state space by clustering together states of the transition matrix that behave similarly. We adapt the soft state aggregation (SSA) mechanism of Duan et al. (2019), to replace the transition matrix in line 6. SSA provides a probabilistic, model-based refinement of the classical idea: instead of assigning each state to a single cluster, it associates each state with multiple meta states through a membership distribution. The transition matrices are then expressed in terms of these meta states of a much smaller rank.

The soft aggregation model is parameterized by $d = |V|$ [8]aggregation distributions and $q$ disaggregation distributions, where $d$ is the total number of states (i.e., discrete loss value) in a Markov chain and $q \ll d$ is the number of (latent) meta states. The model is parameterized by an aggregation map $U \in \mathbb{R}^{d \times q}$, where each row specifies how a raw state distributes its probability mass over the $q$ meta-states, and a disaggregation map $V \in \mathbb{R}^{q \times d}$, where each row gives a distribution from a meta-state back to the original states. Under this formulation, the transition kernel admits the approximation

$$P \approx UV^\top, \quad \text{where} \quad U\mathbf{1}_q = \mathbf{1}_d, \quad \text{and} \quad V^\top \mathbf{1}_d = \mathbf{1}_q.$$

In the above formula, $\mathbf{1}_p, \mathbf{1}_q$ stands for an all one vector of dimension $d$ and $q$, respectively.

---

**Algorithm 5** Soft State Aggregation for Transition Matrix (Duan et al., 2019)

---

**Require:** Empirical count matrix $\mathbf{N}$, number of meta-states $q$, threshold $\delta_0 = 10^{-6}$.

1: Estimate the matrix $\widehat{V}$.

  (i) Form $\widetilde{\mathbf{N}} = \mathbf{N}[\text{diag}(\mathbf{N}^\top \mathbf{1}_q)]^{-1/2}$. Apply truncated singular value decomposition (Simon & Zha, 2000) on $\widetilde{\mathbf{N}}$ and extract the top $q$ right singular vectors $\widehat{h}_1, \ldots, \widehat{h}_q$. Construct $\widehat{D} = [\text{diag}(\widehat{h}_1)]^{-1}[\widehat{h}_2, \ldots, \widehat{h}_q] \in \mathbb{R}^{d \times (q-1)}$. $\qquad \triangleright \Theta(d^2 q)$.

  (ii) Apply a vertex-finding algorithm to the rows of $\widehat{D}$ to obtain $q$ extreme points $\widehat{b}_1, \ldots, \widehat{b}_q$.
$\qquad\qquad\qquad \triangleright \Theta(dq^4)$ via succesive projection algorithm (SPA) (Araújo et al., 2001).

  (iii) For each $1 \le j \le d$, compute

$$\widehat{w}_j^* = \arg\min_{r \in \mathbb{R}^q} \left\| \widehat{d}_j - \sum_{k=1}^q r_k \widehat{b}_k \right\|_2^2 + \left( 1 - \sum_{k=1}^d r_k \right)^2.$$

Clip negative entries to 0, renormalize to unit $\ell_1$-norm, and denote the result by $\widehat{w}_j$. Let $\widehat{W} = [\widehat{w}_1, \ldots, \widehat{w}_q]^\top \in \mathbb{R}^{d \times q}$. Compute $[\text{diag}(\widehat{h}_1)][\text{diag}(\mathbf{N}^\top \mathbf{1}_q)]^{1/2}\widehat{W}$, normalize each column to have unit $\ell_1$-norm, let the resulting matrix be $\widehat{V}$. $\quad \triangleright \Theta(dq^3)$ via Least Square.

2: Estimate the matrix $\widehat{U}$.

  Form empirical transition probability matrix $\widehat{P} = [\text{diag}(\mathbf{N}\mathbf{1}_q)]^{-1}\mathbf{N}$ and compute

$$\widehat{U} = \widehat{P}\widehat{V}(\widehat{V}^\top \widehat{V})^{-1}.$$

$$\triangleright \Theta(d^2 q).$$

3: **return** matrices $\widehat{V}, \widehat{U}$.

---

Next, we outline the intuition of algorithm 5 for estimating the factors $U$ and $V$ from the empirical state–transition counts $\mathbf{N}$. This method has two stages: 1) apply spectral methods to extract the top $d$ left and right singular vectors. 2) introduce a linear reparameterization that transforms these singular vectors into consistent estimators of $U$ and $V$. The resulting estimates yield an approximate transition model that integrates directly into the downstream components of T-TAMER.

*Remark.* Our implementation differs from Duan et al. (2019) in two ways: we do not compute the full SVD in Step (i), and we use successive projection instead of vertex hunting in Step (ii). Both choices reduce the overall time complexity. Diagonal-matrix operations merely perform row- or column-wise scaling and therefore do not incur the cost of full matrix multiplications.

---

[8]For simplicity of presentation, we reuse the symbol $V$ to represent both the loss-value support and the disaggregation matrix; the distinction should be clear from context.

## D.3 Experiment Setup

**Experimental Setup**. Our testbed consists of a single server equipped with two Intel Xeon Gold 6438M processors (3.90 GHz), 512 GB of DDR5 memory, and one NVIDIA RTX 4000 Ada GPU. The server runs Ubuntu 22.04 with Linux kernel 5.15, CUDA version 13.0, and Python 3.10.

**Dataset**. The experimental traces employed in this work are sourced from Dai et al. (2024), where data were collected on dedicated servers equipped with NVIDIA RTX A6000 GPUs (48GB memory), AMD EPYC 7543P 32-Core CPUs, and 256GB DDR4 RAM. The hardware configuration of our own experiments is specified in the main text. The model specifics and data characteristics of EE workloads are specified in Table. 3.

| Model | Dataset |
|---|---|
| **Vision Models** 
 VGG-11,13,16 | Object classification on 8 one-hour urban videos (Agarwal & Netravali, 2023; Hsieh et al., 2018), sampled at 30 fps across day/night. Data are shuffled to discard temporal order and treated as independent images. |
| **Language Models** 
 BERT-base 
 GPT2-medium | Sentiment analysis on Amazon product reviews (McAuley & Leskovec, 2013) and IMDB movie reviews (Pal et al., 2020). |

Table 3: Model Specifics and Dataset Characteristics of EE models.

**Hyperparameters for Pareto Front**. In our experiments for plotting the Pareto Front, we quantize the stochastic loss component using an additive discretization with step size $0.1$ from $[0, 1]$, and estimate the corresponding transition matrix from the induced discrete loss support. The cost term is retained in its continuous form and integrated directly into the objective. Furthermore, accuracy is evaluated on the same dataset for fitting the dynamics. To construct the Pareto frontier, we perform a logarithmic grid search over a range of $\lambda$ values and evaluate the resulting accuracy–latency trade-offs, yielding an empirical approximation of the frontier.

**Hyperparameters for Benchmark Comparison**. For benchmark comparisons with single-threshold and patience-based baselines, we discretize the stochastic loss component using an additive scheme with step size $0.001$ over $[0, 1]$, and estimate the corresponding transition matrix from the induced discrete loss trajectory. We adopt an 80/20 split: 80% of the data is randomly selected for fitting the T-tamer and the benchmarks, and the remaining 20% is used for generating the plot. All other settings follow the previously specified hyperparameters of the Pareto frontier.

**Hyperparameters for SSA Approximation**. For the SSA implementation, we use the version adapted from Duan et al. (2019). For SSA augmented T-tamer, we replace the matrix multiplication step in the T-tamer fitting with two low-rank matrices. The product is computed in two stages: we first compute $G := V^\top \theta$, and then compute $UG$. More details are in App. D.2.

We discretize the stochastic loss component using an additive scheme with step size $0.001$ over the interval $[0, 1]$, and we estimate the corresponding transition matrix from the induced discrete loss trajectory. We use an 80/20 split, as described previously.

The SSA hyperparameter is the number of meta-states $q$. Under a fixed latency requirement, we fit SSA augmented T-tamer with $q \in \{50, 100, \dots, 800\}$. To quantify the estimation quality, we use the $\ell_\infty$ norm between the estimated transition matrix and the true transition matrix as the x-axis. This error may exceed 2 because the estimated matrix is not necessarily row-stochastic.

**Hyperparameters for If-Stop Matrix**. In our experiment for visualizing the if-stop matrix, we quantize the stochastic loss component using an additive discretization with step size $0.1$ from $[0, 1]$, and estimate the corresponding transition matrix from the induced discrete loss support. The remaining distribution-related hyperparameters are specified in the subtitle of Fig. 10.

### D.4 MORE DETAILS ON PARETO FRONT

In Figure 9, we also evaluate the dynamic index strategy on NLP classification tasks using the IMDB (Pal et al., 2020) and Amazon Review (McAuley & Leskovec, 2013) datasets, with BERT-base (Devlin et al., 2019a) and GPT-2 (Radford et al., 2019) as backbone models for EE.

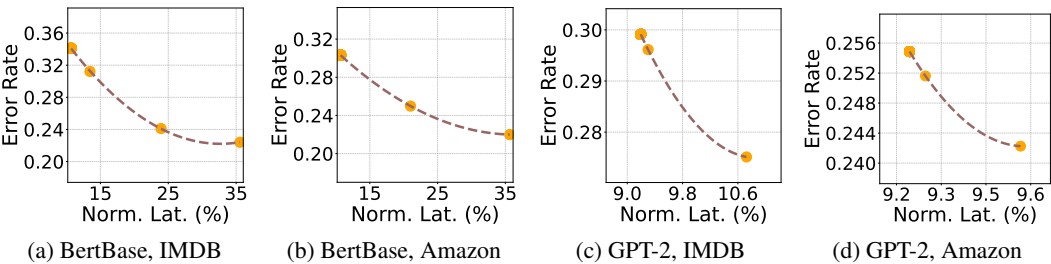

(a) BertBase, IMDB     (b) BertBase, Amazon     (c) GPT-2, IMDB     (d) GPT-2, Amazon

Figure 9: **Pareto Frontier for Language Models.** The frontier shows a sharp accuracy–latency trade-off region, where latency is reduced by up to 90%.

### D.5 VISUALIZING IF-STOP MATRIX

The purpose of our synthetic experiments is to demonstrate that confidence thresholding strategies are not optimal for general distributions. Specifically, we visualize the if-stop matrices under various distributions to validate this claim. More specifically, the loss sequences are generated by independently sampling each loss from several independent distributions over $[0, 1]$, with the latency cost at each ramp fixed to $0.1$ milliseconds.

Here, the purpose is not to emulate practical workloads—where ramp losses are typically positively correlated—but rather to illustrate the general structure of the optimal strategy through the analysis of the resulting if-stop matrices. As shown in the figure, the optimal policy does not coincide with any fixed thresholding rule that is oblivious to the current ramp state. Instead, the decision to stop or continue depends jointly on the realized losses at earlier ramps and the minimum loss across inspected nodes. This highlights the inherent limitation of threshold-based strategies, which, by construction, ignore such dependencies and are therefore provably suboptimal in the general setting.

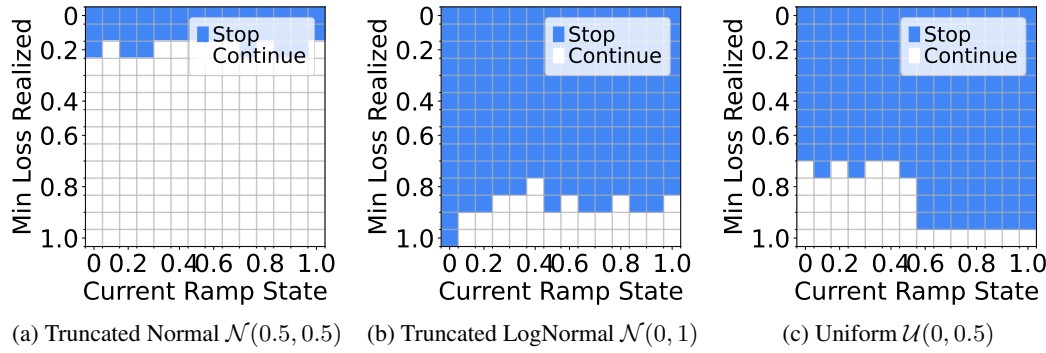

(a) Truncated Normal $\mathcal{N}(0.5, 0.5)$     (b) Truncated LogNormal $\mathcal{N}(0, 1)$     (c) Uniform $\mathcal{U}(0, 0.5)$

Figure 10: Visualization of the If Stop matrix for Synthetic Data

