# OpenReview forum: "T-TAMER: Provably Taming Trade-offs in ML Serving"
_ICLR.cc/2026/Conference — ICLR 2026 Poster_

### Official Review · Reviewer_kRkc · 2025-10-19

**Soundness:** 3
**Presentation:** 3
**Contribution:** 3
**Rating:** 4
**Confidence:** 2

**Summary:**

The paper introduces a theoretical framework to tackle trade-offs in ML serving when there are two objectives. One of the main claims is that it is impossible to achieve offline optimal performance without recall. Then the authors introduce a dynamic indexing strategy to tackle this challenge.

**Strengths:**

1. The writing is clear. The authors abstract out the problem as a Markovian Costly Exploration and offered a principled solution to tackle this tradeoff.

2. The proposed solution also contains optimality guarantees.

**Weaknesses:**

1. It's not clear how one would attain such a loss distribution for complex data in practice in the proposed Markovian setting, the cost associated with it, and whether there is a cheap surrogate that can estimate this loss distribution.

2. This concern remains when the authors attempt to solve this via the dynamic indexing strategy. It is not clear whether discretization/quantization effort would be practical.

3. No recall is a weak baseline. What about heuristics-based approaches that don't have the theoretical optimality guarantees and does the proposed method achieve a significant improvement over them?

Minor issues:
Line 80: There is an extra precedence in the sentence.

**Questions:**

Please see my points above.

In addition, would it be possible to adapt the current framework to account for hard SLO constraints (such as serving latency / throughput) that are pervasive in real-world deployments?

---

> ### Author Response · Authors · 2025-11-27
> **Author Rebuttal (1/2)**
>
> We thank the reviewer for the positive feedback, noting our clear presentation, principled Markovian formulation of the two-objective serving trade-off, and dynamic indexing strategy with optimality guarantees. We also note that certain questions bundle several distinct issues, which we address separately for clarity. We respond to all questions and identify weaknesses below, and are happy to clarify any further points.
>
> ---
>
> > **Weakness 1**: It's not clear how one would attain such a loss distribution for complex data in practice in the proposed Markovian setting, the cost associated with it, and whether there is a cheap surrogate that can estimate this loss distribution.
>
> **Loss Estimation Details**: We thank the reviewer for asking this question.
>
> 1.  Regarding the **loss distribution estimation**, we briefly explain how it is handled in our setting and note that the same procedure is standard across cascaded inference workloads. In our model, the loss distribution is estimated from **softmax-based statistics** (e.g., confidence, entropy) aggregated over a dataset, under the assumption that the workload operates on data drawn from a similar distribution. In practice, this is typically obtained from historical losses on related tasks during the offline stage. When no prior data are available, a simple bootstrap strategy suffices—for example, using the first 20% of incoming samples to estimate the loss distribution and applying the resulting thresholds to the remaining 80%.
>
> 2. Regarding the **estimation of the fixed loss**, for latency measurements, the ML community provides many excellent open-source FLOP profilers. We use a profiler adapted from the lightweight and widely trusted [DeepSpeed FLOP profiler](https://www.deepspeed.ai/tutorials/flops-profiler/). This profiler runs on the CPU side and does not interfere with GPU computation. Importantly, these tools integrate seamlessly with all cascaded inference models studied in this paper, ensuring that our latency estimation remains practical and accurate. As for **other forms of cost**—such as the monetary cost of invoking a particular submodel—these **are typically known a priori**. For instance, the API pricing of models such as GPT is publicly available and can be directly incorporated into the analysis.
>
>  **Cost of Deploying T-TAMER**: In terms of deployment cost, T-tamer is designed to impose **minimal overhead on the online inference pipeline**. All policy learning occurs entirely offline, allowing the computational burden to be shifted away from latency-critical execution.
>
> 1. **T-TAMER executes most of its computation offline, ensuring that user-facing latency remains largely unaffected**. The major offline computations ( i.e., the FLOP estimator and the policy-learning procedure) can be executed entirely on the CPU, leaving GPU-side intermediate-layer inference undisturbed. In many model-serving scenarios, where fast online inference is essential and additional offline computation is acceptable, this redistribution of workload is desirable. We view this clean separation between offline and online computation, enabled by our indexing policy, as a core strength of the approach.
>
> 2. **The inference overhead introduced by T-TAMER is minimal**. At runtime, the only extra cost comes from lightweight communication between the CPU controller and the GPUs to enact the T-TAMER policy. The policy itself reduces to a simple table lookup, whose cost is negligible compared with the model’s actual compute workload (e.g., generating hidden states). Thus, the coordination is **minimal and non-blocking**, and amounts only to standard overhead. Thus, T-tamer introduces negligible overhead on the critical path during inference.
>
> **Cheap surrogate for estimating the loss distribution**: To provide a lightweight surrogate, our revised version introduces **soft state aggregation** [3], which offers a much cheaper approximation for estimating the loss distribution. In T-tamer, loss estimation reduces to learning the transition matrix for each edge and the distribution of the starting vertex, with the dominant cost—both computational and memory—arising from estimating these transition matrices. **Soft state aggregation groups similar states (distinct loss value) to reduce the effective state space, which in turn lowers memory cost and accelerates policy computation**.  This reduces runtime from $O(d^\omega)$ to $O(d^{2} q^{\omega-2})$ and memory from $O(d^{2})$ to $O(dq)$, where $\omega$ is the matrix-multiplication exponent. Such reductions make the method substantially more practical for systems workloads. For additional details, please refer to our global response.

---

> ### Author Response · Authors · 2025-11-27
> **Author Rebuttal (2/2)**
>
> > **Weakness 2**: This concern remains when the authors attempt to solve this via the dynamic indexing strategy. It is not clear whether discretization/quantization effort would be practical.
>
> This is an excellent point, and we thank the reviewer for highlighting it. **Soft state aggregation enables T-TAMER to operate under fine-grained discretization, substantially improving its practical deployability**. We agree that the discretization used in the submitted version was not the most appropriate choice. In additional experiments conducted during the rebuttal period, we adopted state aggregation methods [3] to obtain a more principled quantization scheme. Concretely, we start from a fine-grained discretization (e.g., step size 0.001), and the aggregation procedure clusters these finely discretized values into a smaller set of representative states while controlling the overall approximation error. Please also refer to our response to the previous question and the global response for further details.
>
> ---
>
> > **Weakness 3**:  No recall is a weak baseline. What about heuristics-based approaches that don't have the theoretical optimality guarantees and does the proposed method achieve a significant improvement over them?
>
> - **Added Benchmarks**. In the revised experiments, we include **two additional baselines** (see global response): a **single-threshold method** [1], which applies one fixed confidence threshold across all submodels, and a **patience-based rule** [2], which exits once $K$ consecutive submodels produce identical predictions. For the Pareto Front of the GPT workload, T-tamer **consistently outperforms** the patience-based rule, and it achieves competitive performance with the single-threshold strategy—surpassing it in certain operating regimes.
>
> ---
>
> > **Question 1**: Would it be possible to adapt the current framework to account for hard SLO constraints that are pervasive in real-world deployments?
>
> We thank the reviewer for highlighting this point. When the SLO specifies a hard latency requirement, we can directly perform a **binary search** over $\lambda \in [0,1]$ to select the policy whose latency satisfies the required bound. This procedure invokes T-TAMER only a constant number of times, ensuring negligible additional overhead.
>
> _[1]: Xin, J., Tang, R., Lee, J., Yu, Y., & Lin, J. (2020). DeeBERT: Dynamic early exiting for accelerating BERT inference. arXiv preprint arXiv:2004.12993._
>
> _[2]: Zhou, W., Xu, C., Ge, T., McAuley, J., Xu, K., & Wei, F. (2020). Bert loses patience: Fast and robust inference with early exit. Advances in Neural Information Processing Systems, 33, 18330-18341._
>
> _[3] Duan, Y., Ke, T., & Wang, M. (2019). State aggregation learning from Markov transition data. Advances in Neural Information Processing Systems, 32._

---

### Official Review · Reviewer_m2Vw · 2025-11-01

**Soundness:** 3
**Presentation:** 3
**Contribution:** 3
**Rating:** 6
**Confidence:** 3

**Summary:**

The paper studies accuracy, latency (more generally, bi-objective) trade-offs in cascaded / early-exit ML serving. It models routing + stopping over a DAG and shows a clean result: policies without recall cannot achieve any constant-factor approximation, while recall-based policies can achieve the optimal trade-off in polynomial time. The framework is instantiated as T-TAMER and applied to three common DAGs (line, transitive closure of a line, and directed tree), and experiments on synthetic data plus vision/NLP early-exit workloads show the expected accuracy–latency frontiers.

**Strengths:**

- Clear formalization of cascaded inference as costly exploration over DAGs (line, transitive closure, tree).
- Strong, easy-to-communicate message: no-recall is information-theoretically too weak; recall fixes it.
- Algorithm has polynomial-time preprocessing and 𝑂(𝑛) per-query inference, so it’s not just theory.

**Weaknesses:**

- Experiments are synthetic + standard EE workloads; no real production-style serving stack.
- Some assumptions (Markovian losses, known dists) could be made more operational for systems people.

**Questions:**

1. Can you show one setting where recall is not implementable and quantify the loss?
2. How robust is the policy to mild mis-specification of the loss distributions?

---

> ### Author Response · Authors · 2025-11-27
> **Author Rebuttal (1/1)**
>
> Thank you for your comments regarding the clarity of our formulation and results. We also appreciate your observation about efficiency—our algorithm performs well in practice, demonstrating that its advantages extend beyond theory. Below, we address your questions and concerns related to the presentation, empirical behavior, and robustness of our method.
>
> ---
> > **Weakness 1**: Experiments are synthetic + standard EE workloads; no real production-style serving stack.
>
> Evaluating our strategy on a production-level serving stack would require substantially more GPUs and engineering infrastructure than we currently have access to. If the reviewer has suggestions for lightweight ways to approximate such a setup, we would be more than happy to explore them.
>
> ---
>
> > **Weakness 2**: Some assumptions (Markovian losses, known dists) could be made more operational for systems people.
>
> Thank you very much for bringing this to our attention. We now describe these assumptions in a more direct and easy-to-follow manner. We would be glad to answer any additional questions the reviewer may have about the terminology.
>
> - **Markovian Loss**: Markovian means that in a model arranged in a line ( e.g., A → B → C → D), each model depends only on the one right before it. For example, if we already saw the loss at B, then to understand C or D, we only need this loss at B. We do not need to look back at what happened at A.
>
> - **Known Distribution**: This means before running our method, we first run a similar workload, record how each submodel behaves (like latency, accuracy, or errors), and then fit a distribution to these variables. In other words, it means two things:
>   - The losses do not behave randomly without any pattern.
>   - We have a rough picture of “what the loss usually looks like” for each model stage because we saw similar data before.
>
> ---
>
> > **Question 1**: Can you show one setting where recall is not implementable and quantify the loss?
>
> Yes. In the revised version, we added the benchmark comparison. In this setting, the single threshold strategy represents the case where the recall-based T tamer policy is not implementable at deployment time, which forces the system to fall back to a single threshold. As shown in Figure 5(a), over the GPT 2 model on the IMDB workload, the single threshold strategy can reach at most 92 percent accuracy, while the (recall-based) T tamer policy achieves at most 95 percent. These values indicate the maximum attainable performance of the two strategies under identical conditions and highlight the accuracy gap when recall cannot be implemented. For the Berbase workload, the gap is even larger, reaching 5 percentage points, which further demonstrates the performance degradation caused by the inability to deploy recall.
>
> ---
>
> > **Question 2**: How robust is the policy to mild mis-specification of the loss distributions?
>
> We appreciate this insightful question.
>
> - **Theoretically, our strategy remains stable under small perturbations**. The error will be upper bounded by $\epsilon \times L$, where $\epsilon$ is the misspecification error and $L$ is the length of the longest path of the DAG ($L$ is generally smaller than $n$). Specifically, since the value function is linear in the loss distribution, the sub-optimality of a single step's decision will always be linear in the mis-specification error $\epsilon$. The crucial question is how these errors will accumulate during the dynamic policy. Since the optimal policy relies on dynamic programming (DP), and the next round's reward in the DP formulation appears additively in the current round's reward, these two together determine that the next round's reward will accumulate additively. Hence, the $\epsilon$ errors will accumulate additively for at most $L$ times, where $L$ is the longest path in the dynamic reasoning chain, which is precisely the length of the longest path of the DAG.
>
> - **Our method remains robust under moderate levels of mild transition-matrix misspecification**. In the newly added experiments on soft state aggregation (see our global response), we further analyze how the approximation error introduced by SSA propagates under small misspecifications of the transition matrix. As discussed there, the resulting $\ell_\infty$ matrix-distance deviation between the SSA-estimated transition matrix $T_{\mathrm{SSA}}$ and the true matrix $T$, even when it remains below a constant such as $1$, still leads to a meaningful and reliable accuracy guarantee.

---

### Official Review · Reviewer_nk36 · 2025-11-01

**Soundness:** 2
**Presentation:** 3
**Contribution:** 2
**Rating:** 4
**Confidence:** 2

**Summary:**

This paper presents a theoretical framework designed to optimize bi-objective trade-offs in cascaded inference. The work includes a powerful information-theoretic proof demonstrating that common no-recall confidence-thresholding heuristics are fundamentally suboptimal. The central solution is the Dynamic Indexing Strategy, which the authors prove is both polynomial-time computable and provably optimal for making adaptive inference decisions across various DAG structures.

**Strengths:**

1.	The paper provides theoretical foundations, particularly through its information-theoretic impossibility result for no-recall strategies.
2.	By abstracting cascaded inferences as costly explorations over DAGs, the framework naturally captures diverse topologies (from linear cascades to tree structures).
3.	The work delivers strong theoretical guarantees through its dynamic indexing strategy, proving polynomial-time optimality for multiple DAG structures.

**Weaknesses:**

1. The experimental evaluation is insufficient. It lacks comparison against critical baselines (e.g., standard thresholding, other learned routers) on the same Pareto frontier plots. Furthermore, the practical implementation and advantage of the core DAG-based routing mechanism remain unclear.
2. The theoretical guarantees presented in the paper appear to rely on strong assumptions, most notably the Markov property of the loss sequences. Could you please discuss how these assumptions might influence the experimental results and their generalizability?
3. Although inference is fast, the preprocessing time for the dynamic indexing policy can be computationally expensive. Could you explain how to solve the system with numerous sub-models (large $n$) requiring fine-grained discretization (large $|V|$).
4. There are some instances of missing punctuation. For example, the first paragraph and entry with "metrics" in Section 6 lacks necessary punctuation marks.

**Questions:**

I would appreciate the authors’ responses to the four weaknesses outlined above.

---

> ### Author Response · Authors · 2025-11-27
> **Author Rebuttal (1/2)**
>
> Thank you for your thoughtful and constructive review, and for highlighting the importance of recall, our theoretical strengths, and the generality of our submodel-topology framework. Below, we clarify the implementation details of T-TAMER and its applicability to fine-grained, many-model settings. **Following your insightful suggestions, the revised version adds a new benchmark and incorporates a state-aggregation [3] technique to further accelerate the fine-grained regime**. We sincerely appreciate your feedback and are happy to address any further questions.
>
> ---
>
> > **Weakness 1**: The experimental evaluation is insufficient. It lacks comparison against critical baselines (e.g., standard thresholding, other learned routers) on the same Pareto frontier plots. Furthermore, the practical implementation and advantage of the core DAG-based routing mechanism remain unclear.
>
> **Lack of Experiments and Baselines**. We included **two additional baselines** in the updated experiments (see our global response for details). The first is a single-threshold strategy [1], which applies a uniform confidence threshold to every submodel. The second is a patience-based method [2], which triggers an exit once $K$ successive submodels yield identical predictions.
>
> **Implementation Details of T-TAMER**. In our experiments, the loss range is quantized/discretized using a 0.1 interval, and the transition matrix is inferred from this discretized grid (this applies only to stochastic losses; the cost loss is kept in its continuous form). To visualize the Pareto frontier, we enumerate over $\lambda$ using a grid search to generate a spectrum of accuracy–latency trade-offs. We also incorporate a state-aggregation [3] technique to accelerate T-TAMER. Please see our response to your question on the fine-grained model setting for further details.
>
> **Practical Advantage of DAG-based routing**. Our work initiates a principled step toward moving beyond heuristic routing by adopting a structured formulation. By making the DAG constraints explicit and using a lightweight Markov-style proxy, we obtain a **globally consistent, model-agnostic** routing mechanism that is **broadly applicable** across heterogeneous inference pipelines.
>
> 1.  **DAG-based routing as an early, principled alternative to heuristics**. Our framework takes an early but principled step toward replacing heuristic routing with a structured, globally optimized formulation. Given the rapid shift of modern inference pipelines—from monolithic models to heterogeneous cascades, RL-based controllers, and large-scale MoE systems—a plug-in, model-agnostic routing mechanism becomes increasingly valuable. Heuristic rules often fail to keep pace with these architectural changes, while our formulation naturally adapts without requiring any co-training with the submodels.
>
> 2.  **Theoretical guarantees that match the structural insight**. Beyond the modeling contribution, the theoretical analysis provides rigorous optimality guarantees under the structural assumptions. This form of provable, globally correct routing is absent in existing heuristic-based pipelines and has been consistently appreciated in related literature on structured inference (e.g., optimal decision processes, constrained model composition).
>
> 3. **DAG structure grounded in practical observations**. The DAG-based precedence constraints naturally appear in many common neural network designs, such as early-exit models, staged classifiers, and cascaded inference pipelines. However, these DAG dependencies are usually handled through **local** heuristics, rather than being treated as part of the model’s formal structure or optimization objective. This gap motivates our approach: by making the underlying DAG explicit and treating it as a core structural component, we can optimize routing in a **globally** consistent and principled way.
>
>
> _[1]: Xin, J., Tang, R., Lee, J., Yu, Y., & Lin, J. (2020). DeeBERT: Dynamic early exiting for accelerating BERT inference. arXiv preprint arXiv:2004.12993._
>
> _[2]: Zhou, W., Xu, C., Ge, T., McAuley, J., Xu, K., & Wei, F. (2020). Bert loses patience: Fast and robust inference with early exit. Advances in Neural Information Processing Systems, 33, 18330-18341._
>
>
> _[3] Duan, Y., Ke, T., & Wang, M. (2019). State aggregation learning from Markov transition data. Advances in Neural Information Processing Systems, 32._

---

> ### Author Response · Authors · 2025-11-27
> **Author Rebuttal (2/2)**
>
> > **Weakness 2**: The theoretical guarantees presented in the paper appear to rely on the Markov property of the loss sequences. Could you please discuss how these assumptions might influence the experimental results and their generalizability?
>
> We appreciate the reviewer for raising this important point.
>
> 1.  **Optimality holds for arbitrary correlations in the directed-line case**. As clarified in the global response, the dynamic indexing strategy remains **theoretically optimal** in the directed-line setting even under general (non-Markovian) correlation. The only modification lies in the implementation: instead of propagating the future loss distribution through a transition matrix—as done under the Markov assumption—we directly maintain the empirical joint distribution of losses and compute the index for each state via Bayesian conditioning.
>
> 2. **Markovian remains a good proxy for the EE workload**. Empirically, our experimental results (as stated in the global response) suggest that applying the Markovian correlation model serves as a sufficiently accurate proxy for the underlying correlation, which explains why the method generalizes well despite the stronger theoretical assumptions.
>
> 3. **Generalibility to Other Correlations**. The algorithm remains applicable in non-Markovian environments. It operates broadly and requires no modification under arbitrarily correlated loss sequences for the DAG topologies we consider.
>
> ---
>
> > **Weakness 3**: Although inference is fast, the preprocessing time for the dynamic indexing policy can be computationally expensive. Could you explain how to solve the system with numerous sub-models (large $n$) requiring fine-grained discretization (large $|V|$) .
>
> Thank you for the insightful question. In the fine-grained setting, the algorithm does run more slowly, primarily due to the heavier matrix-multiplication step introduced by the finer discretization. However, we find that **inexpensive surrogate (i.e., state aggregation) approximations mitigate this overhead effectively**. We direct the reviewer to our global response for the supporting experimental results. Below, we provide further details on our experimental setup and how we address this issue:
>
> - **Discretization Granuality**. On consumer-grade GPUs, General Matrix Multiplications (GEMMs) with sizes $N = 1024 \to 2048$ remain highly efficient, so this resolution is already **sufficient for typical deployment**. In our revised experiments, we adopt a discretization level of $\epsilon = 0.001$, under which the algorithm finishes in well under one minute. If further speedup is desired, state-aggregation techniques offer an effective option. Please refer to our global response for supporting experiments.
>
> - **Number of Submodels**. For $\leq 10$ submodels, as in our current experiment setting, storing the exact transition matrix is well within the capabilities of modern hardware. When scaling to around 100 submodels, we suggest adopting the **state-aggregation** method described in the revised manuscript, which we found to yield substantial reductions in both runtime and memory usage.
>
> - **A Note on Sample Efficiency**. **Our approach scales more efficiently than heuristic alternatives on large datasets**, in that T-TAMER estimates all transition matrices in a single pass, whereas heuristics require multiple full scans.
>
> ---
>
> > **Weakness 4**: There are some instances of missing punctuation. For example, the first paragraph and entry with "metrics" in Section 6 lacks necessary punctuation marks.
>
> Thank you for catching this. We have corrected it in the revised version.

---

### Official Review · Reviewer_nhrK · 2025-11-01

**Soundness:** 3
**Presentation:** 3
**Contribution:** 3
**Rating:** 8
**Confidence:** 3

**Summary:**

This paper explores balancing accuracy, latency, and cost in serving large ML models. Traditional cascaded inference runs models from simple to complex, exiting early for easy queries. The authors point out a flaw: once a complex model is used, systems must accept its output (“no-recall”). They propose T-Tamer, a framework viewing inference as a multi-stage decision process. Its key insight: allowing a “with-recall” strategy—choosing the best output at any stage—achieves an optimal trade-off.

**Strengths:**

S1. Original Problem Formulation:
This paper's most original contribution is the critical distinction between "no-recall" and "with-recall" strategies, which re-frames how efficiency trade-offs are understood and optimized.

S2. Theoretical Contributions:
The work provides strong theoretical guarantees, including an information-theoretic proof that "no-recall" policies are inherently suboptimal (cannot achieve any constant‑factor approximation to the offline optimal), and the development of a provably optimal dynamic indexing strategy for "with-recall" settings, which extends efficiently to complex DAG structures.

S3. Generality and Empirical Validation:
The T-TAMER framework is a general, model-agnostic "plug-in" solution, and its practical effectiveness is thoroughly validated through empirical experiments on CV/NLP benchmarks, demonstrating significant latency reductions with minimal accuracy loss.

**Weaknesses:**

W1. Theory–Reality Gap:
The paper's central claim of "provable optimality" rests on strong assumptions that may not hold in practice. The most critical is the Markov property, which assumes a model's loss at one stage only depends on the loss of the immediately preceding stage. In deep neural networks, dependencies are far more complex and long-range, mediated by high-dimensional hidden states.

W2. Limited Experiments and Baselines:
Although results show better accuracy–latency trade-offs, comparisons are mostly against weak “no-recall” baselines. The paper omits tests against state-of-the-art heuristics like confidence- or entropy-based early exits and evaluates only simple linear cascades, leaving complex DAG performance unverified.

**Questions:**

Q1. From what I gather, deploying T‑Tamer requires estimating the loss distributions Dᵢ and transition matrices Pᵢ  from a limited dataset, as well as discretizing the continuous loss space. These implementation steps appear essential for the policy’s effectiveness in practical settings, yet the paper doesn’t seem to discuss them in depth. What are your thoughts on this?

---

> ### Author Response · Authors · 2025-11-27
> **Author Rebuttal (1/2)**
>
> Thank you for your review and for recognizing the importance of recall, our theoretical contributions, and the general, model-agnostic nature of our framework. Below, we address your concerns regarding the Markovian assumption, the implementation details of T-TAMER, and the experiments, and we are happy to answer any further questions you may have.
>
> ---
>
> > **Weakness 1**: **Theory–Reality Gap**: The paper's central claim of "provable optimality" rests on strong assumptions that may not hold in practice. The most critical is the Markov property, which assumes a model's loss at one stage only depends on the loss of the immediately preceding stage. In deep neural networks, dependencies are far more complex and long-range, mediated by high-dimensional hidden states.
>
> We thank the reviewers for this insightful question. While the Markovian assumption does not perfectly capture the true loss correlations in cascaded inference, from our added experiment (please see global rebuttal response), it provides an **effective and tractable** proxy in practice. Importantly, our method is **not restricted to the Markovian assumption** for the single-line case.
>
> 1.  **Optimality on the directed line holds for arbitrary correlation**. For the directed line strategy, the adaptive indexing strategy remains optimal under arbitrary correlation. After revisiting our proof, we found that the optimality result in the directed-line setting also holds under general correlation, with the only change being that the loss distribution must now be propagated via Bayesian conditioning.
>
> 2. **Markovian structure provides a reasonable proxy for arbitrary correlation**, within our early-exit experiments. In our EE model, each ramp is designed to make full use of the information available at its corresponding stage of the backbone. Thus, the **hidden states** on which each ramp makes predictions **exhibit Markovian** behavior. Empirically, T-tamer performs **competitively against standard baselines** —including single-threshold and patience-based strategies—demonstrating that its effectiveness does not depend on the correlation being strictly Markovian.
>
> 3.  **Markovian assumption is essential for theoretical optimality**. This assumption is the key condition that permits **constant-factor approximation bounds**. Without it—even in simpler settings with no precedence constraints—known polynomial-time methods cannot guarantee better than a 4-approximation. Once we drop the Markov property, the problem also loses its simple index-style structure, and the analysis must rely on heavier reduction techniques instead.
>
> ---
>
> > **Weakness 2**: **Limited Experiments and Baselines**: Although results show better accuracy–latency trade-offs, comparisons are mostly against weak “no-recall” baselines. The paper omits tests against state-of-the-art heuristics like confidence- or entropy-based early exits and evaluates only simple linear cascades, leaving complex DAG performance unverified.
>
> 1.  **We have included two additional baselines in the new experiments**, please see our global response. The first is a single-threshold method [1] , which applies one fixed confidence threshold across all submodels. The second is a patience-based approach [2] , which exits once $K$ consecutive submodels output the same prediction.
>
> 2.  We would like to clarify that **confidence-based and entropy-based early-exit rules are special cases of no-recall policies** in the directed-line T-TAMER setting. T-TAMER maintains a stop/no-stop table indexed by each submodel’s loss. When this table assigns “no-stop’’ to all states above a loss threshold:
>   - Setting ` loss = 1 − confidence` recovers the confidence-based rule, and
>   - Setting `loss = entropy` recovers the entropy-based rule, for Shannon entropy.
>
> If there are other benchmarks you would like us to evaluate, we would be happy to include them.
>
> _[1]: Xin, J., Tang, R., Lee, J., Yu, Y., & Lin, J. (2020). DeeBERT: Dynamic early exiting for accelerating BERT inference. arXiv preprint arXiv:2004.12993._
>
> _[2]: Zhou, W., Xu, C., Ge, T., McAuley, J., Xu, K., & Wei, F. (2020). Bert loses patience: Fast and robust inference with early exit. Advances in Neural Information Processing Systems, 33, 18330-18341._

---

> ### Author Response · Authors · 2025-11-27
> **Author Rebuttal (2/2)**
>
> > **Question 1**: From what I gather, deploying T‑Tamer requires estimating the loss distributions Dᵢ and transition matrices Pᵢ from a limited dataset, as well as discretizing the continuous loss space. These implementation steps appear essential for the policy’s effectiveness in practical settings, yet the paper doesn’t seem to discuss them in depth. What are your thoughts on this?
>
> T-tamer’s main challenge is **discretizing a continuous distribution**, a classic problem in MDPs. Its performance typically improves with finer-grained discretization. To achieve strong results, T-tamer therefore relies on a sufficiently fine discretization of the underlying continuous distribution. To offset the associated computational and memory cost, we further accelerate T-tamer fitting using **state-aggregation techniques** [3].
>
> - **Details on Discretization and the Pareto Frontier**. In the submitted experiments, we discretize the loss space with a step size of 0.1 and estimate the transition matrix from this discretization (only for stochastic losses; the cost loss uses its actual value). To plot the Pareto front, we run a grid search over $\lambda$ to obtain different accuracy–latency trade-offs. Our additive discretization with step size~0.1 was intended as a simple baseline and can introduce up to 0.1 error. We now use a finer grid to reduce this error.
>
> - **State Aggregation for More Efficient T-TAMER**. After the initial submission, we adopted a **finer grid and applied \textbf{state-aggregation}**[3]--based factorization to accelerate the most time-consuming step: multiplying two $d \times d$ matrices, for $d$ as the dimension of the transition matrix. Specifically, for each transition matrix $P$, we approximate $$ P \approx A B,\qquad A \in \mathbb{R}^{d \times q},\ B \in \mathbb{R}^{q \times d},\ q \ll d .$$ This decomposition replaces the most expensive step---multiplying two $d \times d$ matrices---with three smaller multiplications involving $d \times q$ and $q \times d$ matrices. Under a fast matrix multiplication algorithm with exponent $\omega$, this reduces runtime from $ O(d^\omega) $ to  $ O(d^2 q^{\omega - 2})$, and decreases the space complexity from $O(d^2)$ to $O(d q)$.
>
>
> _[3]: Duan, Y., Ke, T., & Wang, M. (2019). State aggregation learning from Markov transition data. Advances in Neural Information Processing Systems, 32._

---

### Author Response · Authors · 2025-11-27
**Global Response #1: Additional Experiments with More Benchmarks**

Upon the reviewers’ request, we have added comparisons against relevant benchmarks and are still actively conducting additional evaluations. If the reviewers have suggestions for alternative thresholding strategies, we are very open to incorporating them. The updated experiments include the following benchmarks:

- **Comparison with Confidence-Based approaches**[1]: Single threshold strategy, which use one threshold to govern all submodels, can be implemented efficiently via binary search. On the contrary, we omit comparisons to optimal multi-threshold configuration in that it requires grid search and is hence computationally expensive.

- **Comparison with Patience Based approaches** [2] : The patience-based approach exits once $K$ consecutive submodels output the same prediction.

Results are shown in a table with precision kept to two decimals. Using the **fitted** 1-threshold curve, we estimate the corresponding error rate at matched latency, capping values above 1 at 1.0. **For the NLP workload, T-tamer can achieve 90% accuracy with a** $2\times$ **speedup.** Additional experimental details are provided in the comment below.


_[1]: Xin, J., Tang, R., Lee, J., Yu, Y., & Lin, J. (2020). DeeBERT: Dynamic early exiting for accelerating BERT inference. arXiv preprint arXiv:2004.12993._

_[2]: Zhou, W., Xu, C., Ge, T., McAuley, J., Xu, K., & Wei, F. (2020). Bert loses patience: Fast and robust inference with early exit. Advances in Neural Information Processing Systems, 33, 18330-18341._

---

> ### Author Response · Authors · 2025-11-27
> **GPT workload on IMDB Dataset, Early Exit Models**
>
> The table summarizes how T-Tamer compares with the 1-Threshold baseline across different latency levels. Since the performance of patience-based methods cannot be searched over varying latency requirements, we directly use the case of $K=1$ for comparison.
>
> ---
>
> | Latency (%) | T-Tamer Error Rate | 1-Threshold Error Rate | Difference |
> |:---:|:---:|:---:|:---:|
> | 9.19 | 0.30 | 0.30 | 0.00 |
> | 23.55 | 0.21 | 0.21 | 0.01 |
> | 78.96 | **0.05** | 0.12 | -0.07 |
> | 84.32 | **0.05** | 0.14 | -0.10 |
> | 84.65 | **0.05** | 0.15 | -0.10 |
>
> **Patience-Based Method**: In the same experimental setting, the patience-based method with $K=1$ achieves a latency equal to 49.92% of the backbone model’s and attains an error rate of 0.10 when testing.
>
> ---
> **Analysis**: The table shows that:
>
> - **Low latency** (≈9–24%):
> Both T-Tamer and 1-Threshold have nearly identical errors. We believe the reason is that, in the most constrained regime, both methods behave similarly because the decision space is essentially degenerate.
>
> - **Higher latency** (≈79–85%):
> T-Tamer effectively leverages the extra information and stabilizes at 0.05 error, while 1-Threshold degrades to 0.12–0.15, yielding a consistent **7–10%** improvement.
>
> - **Compared to Patience-based** (49.36% latency, 0.14 error):
> T-Tamer achieves clearly lower errors across **both** lower and higher latencies, essentially dominating this heuristic.
>
> These observations underscore T-Tamer’s superior ability to adapt its waiting strategy as latency increases.

---

> ### Author Response · Authors · 2025-12-02
> **GPT Workload on Amazon-Reviews, Early Exit Models**
>
> The table summarizes how T-Tamer compares with the 1-Threshold baseline across different latency levels. Since the performance of patience-based methods cannot be searched over varying latency requirements, we directly use the case of $K=1$ for comparison.
>
> ---
> | Latency (%) | T-Tamer Error | 1-Threshold Error Rate | Difference |
> |---|---|---|---|
> | 9.19 | 0.25 | 0.24 | 0.01 |
> | 14.24 | 0.22 | 0.20 | 0.01 |
> | 71.10 | **0.04** | 0.10 | -0.05 |
> | 76.25 | **0.04** | 0.12 | -0.08 |
> | 76.71 | **0.04** | 0.12 | -0.08 |
>
> **Patience-Based Method**: In the same experimental setting, the patience-based method with $K=1$ achieves a latency equal to 20.75% of the backbone model’s and attains an error of 0.18 when testing.

---

> ### Author Response · Authors · 2025-12-03
> **BERT Workload on Amazon-Reviews, Early Exit Models**
>
> The table summarizes how T-Tamer compares with the 1-Threshold baseline across different latency levels. Since the performance of patience-based methods cannot be searched over varying latency requirements, we directly use the case of $K=1, 2$ for comparison.
>
> ---
>
> | Latency (%) | T-Tamer Error | 1-Threshold Error Rate | Difference |
> |---|---|---|---|
> | 51.18 | **0.14** | 0.86 | -0.73 |
> | 71.89 | **0.08** | 1.00 | -0.92 |
> | 79.17 | **0.08** | 1.00 | -0.92 |
> | 81.27 | **0.07** | 1.00 | -0.93 |
>
> **Patience-Based Method**: In the same experimental setting, the patience-based method with $K=1$ achieves a latency equal to 26.52% of the backbone model’s and attains an error of 0.22 when testing; Moreover, the patience-based method with $K=2$ achieves a latency equal to 70.23% of the backbone model’s and attains an error of 0.12 when testing.

---

> ### Author Response · Authors · 2025-12-03
> **BERT Workload on IMDB Dataset, Early Exit Models**
>
> The table summarizes how T-Tamer compares with the 1-Threshold baseline across different latency levels. Since the performance of patience-based methods cannot be searched over varying latency requirements, we directly use the case of $K=1, 2$ for comparison.
>
> ---
>
> | Latency (%) | T-Tamer Error | 1-Threshold Error Rate | difference |
> |---|---|---|---|
> | 10.69 | 0.32 | 0.16 | 0.17 |
> | 13.18 | 0.29 | 0.13 | 0.16 |
> | 83.45 | **0.05** | 1.00 | -0.95 |
> | 85.27 | **0.05** | 1.00 | -0.95 |
> | 85.47 | **0.05** | 1.00 | -0.95 |
>
> **Patience-Based Method**: In the same experimental setting, the patience-based method with $K=1$ achieves a latency equal to 28.53% of the backbone model’s and attains an error of 0.27 when testing; Moreover, the patience-based method with $K=2$ achieves a latency equal to 73.22% of the backbone model’s and attains an error of 0.11 when testing.

---

### Author Response · Authors · 2025-11-27
**Global Response #2: Accelerating T-Tamer with Soft State Aggregation**

We incorporate **state aggregation** to enable efficient, fine-grained discretization of T-TAMER. The objective is to construct a compact set of discrete states (i.e., distinct discrete loss valuations) that approximate the underlying continuous Markov process. To integrate SSA, we replace the original transition matrix with two lower-dimensional, low-rank matrices, thereby reducing the offline fitting time of T-TAMER and lowering the space complexity of the transition representation. This method is substantially more efficient than the additive discretization used in the submitted version of the paper.

- **State Aggregation Intro**. Soft state aggregation is a variant of the classical state–aggregation framework in which each
raw state may softly mix across multiple meta-states rather than belonging to a single one.
Formally, the Markov transition matrix admits a nonnegative factorization
$$
P = U V^\top,
$$
where $U$ specifies how each raw state aggregates into the $r$ meta-states and $V$ specifies how
each meta-state disaggregates back to the raw states.

- **Soft State Aggregation Implementation**. More specifically, we adopt the SSA formulation in [1]. The learning algorithm exploits the simplex geometry inherent in the model: each row of $P$ lies in an $(r-1)$-dimensional simplex whose vertices correspond to the meta-states. By projecting the empirical transition matrix, recovering these vertices through a vertex-finding step (we employ a successive-projection method rather than the routine used in the original paper), and representing each state as a convex combination of the recovered corners, the algorithm consistently reconstructs both $U$ and $V$.

- **Soft State Aggregation Result**. Our experiment ( **Figure 6 in the revised PDF**) reports the performance of SSA-augmented T-Tamer as the number of meta states increases. Across CV workloads, the SSA version closely matches the exact T-Tamer, achieving nearly identical accuracy while reducing storage by a factor of approximately $2.5$ to $3$. We also observe that T-Tamer remains highly **stable**, with accuracy varying by at most $\pm 0.02$, even **under the misspecification error** (measured by the $\ell_\infty$ norm) introduced by SSA.

_[1]: Duan, Y., Ke, T., & Wang, M. (2019). State aggregation learning from Markov transition data. Advances in Neural Information Processing Systems, 32._

---

> ### Author Response · Authors · 2025-12-04
> **VGG-13 on Auburn Dataset**
>
> The following tables summarize the performance of SSA-augmented T-Tamer on the VGG-13 Auburn workload under varying latency requirements and different numbers of meta-states.
>
> ---
>
> > **Latency = 1.5 ms**
>
> | #Meta-States | Estimation-Error | Accuracy | Performance-Gap |
> |--------------|------------------|----------|----------------|
> | 101 | 5.950 | 0.936 | 0.005 |
> | 151 | 5.950 | 0.938 | 0.003 |
> | 201 | 3.351 | 0.928 | 0.013 |
> | 251 | 2.932 | 0.915 | 0.026 |
> | 301 | 2.989 | 0.923 | 0.018 |
> | 351 | 1.877 | 0.936 | 0.005 |
> | 401 | 2.924 | 0.938 | 0.003 |
> | 451 | 2.566 | 0.941 | 0.000 |
> | 501 | 0.602 | 0.941 | 0.000 |
> | 551 | 0.549 | 0.941 | 0.000 |
> | 601 | 0.562 | 0.943 | 0.003 |
> | 651 | 0.636 | 0.943 | 0.003 |
> | 701 | 0.712 | 0.941 | 0.000 |
> | 751 | 0.677 | 0.941 | 0.000 |
>
> ---
>
> > **Latency = 2.0 ms**
>
> | #Meta-States | Estimation-Error | Accuracy | Performance-Gap |
> |--------------|------------------|----------|----------------|
> | 101 | 5.279 | 0.936 | 0.003 |
> | 151 | 4.330 | 0.933 | 0.005 |
> | 201 | 3.447 | 0.933 | 0.005 |
> | 251 | 2.834 | 0.936 | 0.003 |
> | 301 | 2.472 | 0.931 | 0.008 |
> | 351 | 1.450 | 0.933 | 0.005 |
> | 401 | 1.204 | 0.936 | 0.003 |
> | 451 | 2.122 | 0.938 | 0.000 |
> | 501 | 0.476 | 0.938 | 0.000 |
> | 551 | 0.649 | 0.938 | 0.000 |
> | 601 | 0.669 | 0.936 | 0.003 |
> | 651 | 0.679 | 0.938 | 0.000 |
> | 701 | 0.747 | 0.938 | 0.000 |
> | 751 | 0.816 | 0.938 | 0.000 |
>
> ---
>
> > **Latency = 2.7 ms**
>
> | #Meta-States | Estimation-Error | Accuracy | Performance-Gap |
> |--------------|------------------|----------|----------------|
> | 101 | 6.160 | 0.951 | 0.000 |
> | 151 | 3.369 | 0.949 | 0.003 |
> | 201 | 3.118 | 0.951 | 0.000 |
> | 251 | 2.949 | 0.946 | 0.005 |
> | 301 | 2.758 | 0.951 | 0.000 |
> | 351 | 2.612 | 0.951 | 0.000 |
> | 401 | 1.292 | 0.951 | 0.000 |
> | 451 | 2.800 | 0.951 | 0.000 |
> | 501 | 1.228 | 0.951 | 0.000 |
> | 551 | 0.943 | 0.951 | 0.000 |
> | 601 | 0.720 | 0.951 | 0.000 |
> | 651 | 0.871 | 0.951 | 0.000 |
> | 701 | 0.709 | 0.951 | 0.000 |
> | 751 | 0.874 | 0.951 | 0.000 |

---

### Author Response · Authors · 2025-11-27
**Global Response # 3: Clarifications on the Markovian Assumption**

We would like to clarify that although the intuition behind T-tamer is motivated by Markovian structure, the **Markovian assumption is not central to our algorithm's practicability**. Instead, the Markovian assumption is invoked solely for establishing **theoretical optimality** guarantees.

- The Markovian structure is required solely for establishing the **theoretical optimality** guarantees on general graph topologies; it does not constrain the practical deployment or execution of our algorithm (except that it is not guaranteed to be theoretically optimal). Even for general graphs, the dynamic index produced by our procedure remains well defined and can be applied directly.

- **Our optimality result in the directed-line setting generalizes to arbitrary correlation**. After a careful revisit of our proof during the rebuttal period, we found that our optimality result in the *directed-line* setting also holds under general correlation. The only change lies in the algorithmic implementation: unlike the Markovian case, we can no longer use a simple transition matrix to propagate the loss distribution. Instead, we update the distribution via **Bayesian conditioning**.

- **Markovian assumption is essential for theoretical optimality**. At a high level, this condition is what makes any constant-factor approximation achievable. If it is removed—even in simpler models without precedence constraints—**no known polynomial-time algorithm can do better than a factor-4 guarantee**. Abandoning the Markov property also eliminates the clean index-type structure of the optimal solution, forcing analyses to depend instead on more involved reduction-based arguments.

- **Markovian approximation remains a good proxy**.
  - For intra-model cascaded inference like EE workloads, vanilla T-Tamer already performs well relative to standard benchmarks (Please see our global response #1), suggesting that the Markovian approximation remains a good proxy for EE workloads.

---

### Author Response · Authors · 2025-12-03
**Author Final Remarks**

We thank the reviewers and Area Chairs for their thorough and insightful feedback. **We have responded to every weakness and question raised by all reviewers and revised the manuscript accordingly**. The corresponding updates in the revised manuscript are shown in color: `blue` indicates newly added paragraphs, and `red` marks paragraphs that have been repositioned.

---

### T-TAMER Contributions:

We briefly summarize our contribution:

- **DAG-based costly inspection w/ theoretical guarantees:** We cast cascade inference as a DAG-structured routing/inspection problem and prove information-theoretic lower bounds showing the inherent suboptimality of all “no-recall’’ policies.
- **Efficient, model-agnostic plug-in framework**. We propose T-TAMER, a model-agnostic plug-in framework whose indexing strategy is optimal on directed lines and Markov-optimal on broader DAG families.  We further incorporate soft state aggregation to support efficient fine-grained implementation with strong robustness to mild estimation noise.
- **Robust performance over practical workloads**. Across CV and NLP, T-TAMER achieves better performance relative to commonly used heuristics, including confidence thresholding and patience-based strategies.

---

### List of Major Changes:

- **Added Experiments** (to address concerns from Reviewer nhrK, nk36, and m2Vw.): We include comparisons against single-thresholding and patience-based thresholds, both of which are standard benchmarks in early-exit evaluation.
  - In Section 6.2, Figure 5(a), we add a benchmark comparison over the GPT-2 model on the IMDB workload.
  - In Section 6.2, Figure 5(b), we add a benchmark comparison over the GPT-2 model on the Amazon Review workload.
  - In Section 6.2, Figure 5(c), we add a benchmark comparison over the Bert-Base model on the IMDB workload.
  - In Section 6.2, Figure 5(d), we add a comparison of the Bert-Base model on the Amazon Review workload.
  - In Section 6.2, Figure 6 (a), (b), (c), (d), we add four experiments on the effect of SSA approximation on accuracy.
  - In Section 6.1, we add detailed introductions and discussions with respect to two added benchmarks.

- **Added SSA Algorithms for Practical Deployment** (to address concerns from Reviewer nhrK, Reviewer nk36, Reviewer m2Vw, and Reviewer kRkc): We incorporate soft state aggregation (SSA) to accelerate the implementation of T-TAMER under fine-grained losses, with the following two experiments.

  - **Number of Meta States vs. Approximation Quality**:  Our results show that T-TAMER remains stable under soft state aggregation, with up to a $5 \times$ reduction in dimensionality having negligible impact on performance.
    - In Section 6.2, Figure 6(c), (d), we added experiments on T-Tamer's accuracy against various numbers of meta states over the VGG-13 model on the Auburn workload, with latency requirements equal to 1.5ms and 2.0 ms.

  - **Estimation Error vs. Approximation Quality**: Our results show that SSA-augmented T-TAMER remains within $\pm 0.02$ of the accuracy of exact T-TAMER (which uses the true transition matrices), regardless of the magnitude of the estimation error, when the number of meta states exceeds 50.
    - In Section 6.2, Figure 6(a), (b), we added experiments on the approximation quality as a function of the estimation error of T-Tamer, over the VGG-13 model on the Auburn workload, with latency requirements equal to 1.5ms and 2.0 ms.

  - **Background and Implementation Details for SSA**.
    - In Section D.2, we added the introduction, background, and implementation details regarding the SSA augmented T-Tamer.

- **Optimality Analysis to Arbitrary Correlation for Directed Line** (to address concerns from Reviewer nhrK and Reviewer nk36):
  - Upon reexamining the proof, we figured out that in the single directed line setting, the generalized indexing strategy is actually theoretically optimal even under arbitrary correlation.
  - In Section 4.3, we added a remark to clarify the generalization to arbitrary correlation.

---

### Global Responses:
In the three global responses, we present a comprehensive set of explanations and evaluations that address several common questions raised by the reviewers.

---

### Individual Responses:
Under specific reviewer feedback, we provide detailed responses to each weakness and question raised by individual reviewers.

---

### Meta-Review · Area_Chair_qPBr · 2025-12-18

**Summary:**

The reviewers agree the paper solves the important problem of trading off accuracy and latency in cascaded / early-exit ML serving with strong theory and empirical validation. There are concerns on the Markov property assumption, experiments, computation cost, and other technical details, but the authors thoroughly address them though local and global responses. In particular it is good to know that the Markovian assumption is not required for the proposed algorithm, but just needed for theoretical optimality. Two reviewers gave high scores, while two others gave a lower score, but with low confidence. Overall, the AC believes there is enough support to accept the paper.

**Reviewer Concerns:**

There are concerns on the Markov property assumption, experiments, computation cost, and other technical details, but the authors thoroughly address them though their responses. The Markovian assumption is not required for the proposed algorithm, but just needed for theoretical optimality. For experiments, two additional baselines are included for comparison. Finally, the computation overhead is reduced using state aggregation approximations. There does not seem to be concerns that are outstanding.

**Reviewer Scores:**

Reviewer nhrK: score is 8;
Reviewer nk36: score is 4, but confidence is 2;
Reviewer m2Vw: score is 6;
Reviewer kRkc: score is 4, but confidence is 2

The AC does not see these scores changing based on the discussions so far.

---

### Decision · Program_Chairs · 2026-01-26

Accept (Poster)